# When to Sense and Control?
# A Time-adaptive Approach for Continuous-Time RL

**Lenart Treven,*Bhavya Sukhija, Yarden As, Florian Dörfler, Andreas Krause**
ETH Zurich, Switzerland

## Abstract

Reinforcement learning (RL) excels in optimizing policies for discrete-time Markov decision processes (MDP). However, various systems are inherently continuous in time, making discrete-time MDPs an inexact modeling choice. In many applications, such as greenhouse control or medical treatments, each interaction (measurement or switching of action) involves manual intervention and thus is inherently costly. Therefore, we generally prefer a time-adaptive approach with fewer interactions with the system. In this work, we formalize an RL framework, *Time-adaptive Control & Sensing* (**TaCoS**), that tackles this challenge by optimizing over policies that besides control predict the duration of its application. Our formulation results in an extended MDP that any standard RL algorithm can solve. We demonstrate that state-of-the-art RL algorithms trained on TaCoS drastically reduce the interaction amount over their discrete-time counterpart while retaining the same or improved performance, and exhibiting robustness over discretization frequency. Finally, we propose OTaCoS, an efficient model-based algorithm for our setting. We show that OTaCoS enjoys sublinear regret for systems with sufficiently smooth dynamics and empirically results in further sample-efficiency gains.

## 1 Introduction

Nearly all state-of-the-art RL algorithms (Schulman et al., 2017; Haarnoja et al., 2018; Lillicrap et al., 2015; Schulman et al., 2015) were developed for discrete-time MDPs. Nevertheless, continuous-time systems are ubiquitous in nature, ranging from robotics, biology, medicine, environment and sustainability etc. (cf. Spong et al., 2006; Jones et al., 2009; Lenhart and Workman, 2007; Panetta and Fister, 2003; Turchetta et al., 2022). Such systems can be naturally modeled with stochastic differential equations (SDEs), but computational approaches necessitate discretization. Furthermore, in many applications, obtaining measurements and switching actions is expensive. For instance, consider a greenhouse of fruits or medical treatment recommendations. In both cases, each measurement (crop inspection, medical exam) or switching of actions (climate control, treatment adjustment) typically involves costly human intervention. Hence, minimizing such interactions with the underlying system is desirable. This underlying challenge is rarely addressed in the RL literature.

In practice, a time-equidistant discretization frequency is set, often manually, adjusted to the underlying system's characteristic time scale. This is challenging, however, especially for unknown/uncertain systems, and systems with multiple dominant time scales (Engquist et al., 2007). Therefore, for many real-world applications having a global frequency of control is inadequate and wasteful. For example, in medicine, patient monitoring often requires higher frequency interaction during the onset of illness and lower frequency interactions as the patient recovers (Kaandorp and Koole, 2007).

In this work, we address this limitation of standard RL methods and propose a novel RL framework, **T**ime-**a**daptive **Co**ntrol & **S**ensing (**TaCoS**). TaCoS reduces a general continuous-time RL problem with underlying SDE dynamics to an equivalent discrete-time MDP, that can be solved with any

---

*Correspondence to `lenart.treven@inf.ethz.ch`

38th Conference on Neural Information Processing Systems (NeurIPS 2024).

RL algorithm, including standard policy gradient methods like PPO and SAC (Schulman et al., 2017; Haarnoja et al., 2018). We summarize our contributions below.

**Contributions**

1. We reformulate the problem of time-adaptive continuous time RL to an equaivalent discrete-time MDP that can be solved with standard RL algorithms.

2. Using our formulation, we extend standard policy gradient techniques (Haarnoja et al. (2018) and Schulman et al. (2017)) to the time-adaptive setting. Our empirical results on standard RL benchmarks (Freeman et al., 2021) show that TACoS outperforms its discrete-time counterpart in terms of policy performance, computational cost, and sample efficiency.

3. To further improve sample efficiency, we propose a model-based RL algorithm, OTACoS. OTACoS uses well-calibrated probabilistic models to capture epistemic uncertainty and, similar to Curi et al. (2020) and Treven et al. (2023), leverages the principle of optimism in the face of uncertainty to guide exploration during learning. We theoretically prove that OTACoS suffers no regret and empirically demonstrate its sample efficiency.

## 2 Problem statement

We consider a general nonlinear continuous time dynamical system with continuous state $\mathcal{X} \subset \mathbb{R}^{d_x}$ and action $\mathcal{U} \subset \mathbb{R}^{d_u}$ space. The underlying dynamics are governed by a (controllable) SDE:

$$d\boldsymbol{x}_t = \boldsymbol{f}^*(\boldsymbol{x}_t, \boldsymbol{u}_t)dt + \boldsymbol{g}^*(\boldsymbol{x}_t, \boldsymbol{u}_t)d\boldsymbol{B}_t. \tag{1}$$

Here $\boldsymbol{x}_t \in \mathcal{X}$ is the state at time $t$, $\boldsymbol{u}_t \in \mathcal{U}$ the control input, $\boldsymbol{f}^*, \boldsymbol{g}^*$ are unknown drift and diffusion functions and $\boldsymbol{B}_t$ is a standard Brownian motion in $\mathbb{R}^{d_B}$. Our goal is to find a control policy $\boldsymbol{\pi}_{\mathcal{U}} : \mathcal{X} \times \mathcal{T} \to \mathcal{U}$ which maximizes an unknown reward $b^*(\boldsymbol{x}_t, \boldsymbol{u}_t)$ over a fixed horizon $\mathcal{T} \overset{\text{def}}{=} [0, T]$, i.e.,

$$\max_{\boldsymbol{\pi} \in \Pi} \mathbb{E}\left[\int_{t \in \mathcal{T}} b^*(\boldsymbol{x}_t, \boldsymbol{\pi}_{\mathcal{U}}(\boldsymbol{x}_t, t))dt\right],$$

where the expectation is taken w.r.t. the policy and stochastic dynamics and $\Pi$ is the class of policies[2] over which we search.

In practice, we can only measure the system state and execute control policies in discrete points in time. In this work, we focus on problems where state measurement and control are synchronized in time. We refer to these synchronized time points as *interactions* in the following parts of this paper. Synchronizing state measurement and control contrasts standard time-adaptive approaches such as event-triggered control (Heemels et al., 2021), where the state is measured arbitrarily high frequency and control inputs are changed only so often to ensure stability. It is also in contrast to the complementary setting, where control inputs are changing at an arbitrarily high frequency but measurements are collected adaptively in time (Treven et al., 2023). An adaptive control approach as Heemels et al. (2021) is very important for many real-world applications but similarly, an adaptive measurement strategy is crucial for efficient learning in RL (Treven et al., 2023). Our approach treats both of these requirements jointly.

We consider two different scenarios for continuous-time control: (*i*) Penalizing interactions with some cost, (*ii*) bounded number of interactions, i.e., hard constraint on control/measurement steps.

**Interaction cost** We consider the setting where every interaction we take has an inherent cost $c(\boldsymbol{x}_t, \boldsymbol{u}_t) > 0$. Note that we consider this cost structure for its simplicity and TACoS works for more general cost functions that depend on the duration of application for the action $\boldsymbol{u}_t$ or the previous action $\boldsymbol{u}_{t-1}$ and thus captures many practical real-world settings. We define this task more formally below

$$\max_{\boldsymbol{\pi} \in \Pi, \pi_{\mathcal{T}}} \mathbb{E}\left[\sum_{i=0}^{K-1} \int_{t_{i-1}}^{t_i} b^*(\boldsymbol{x}_t, \boldsymbol{\pi}_{\mathcal{U}}(\boldsymbol{x}_{t_{i-1}}, t_{i-1}))dt - c(\boldsymbol{x}_{t_{i-1}}, \boldsymbol{\pi}_{\mathcal{U}}(\boldsymbol{x}_{t_{i-1}}, t_{i-1}))\right], \tag{2}$$

$$t_i = \pi_{\mathcal{T}}(\boldsymbol{x}_{t_{i-1}}, t_{i-1}) + t_{i-1}, \ t_0 = 0, t_K = T, \ \forall(\boldsymbol{x}, t) \in \mathcal{X} \times \mathcal{T}; \pi_{\mathcal{T}}(\boldsymbol{x}, t) \in [t_{\min}, t_{\max}].$$

Here $t_{\min} > 0$ is the minimal duration for which we have to apply the control, $t_{\max} \in [t_{\min}, T]$ the maximum duration, and $\pi_{\mathcal{T}}$ is a policy that predicts the duration of applying the action.

---

[2]We assume that $\Pi$ is the set of $L_{\boldsymbol{\pi}}$-Lipschitz policies

**Bounded number of interactions** In this setting, the number of interactions with the system is limited by a known amount $K$. Intuitively, this represents a scenario where we have a finite budget for the inputs that we can apply and have to decide on the best strategy to space these $K$ inputs over the full horizon. A formal definition of this task is given below

$$\max_{\boldsymbol{\pi} \in \Pi, \pi_{\mathcal{T}}} \mathbb{E} \left[ \sum_{i=0}^{K-1} \int_{t_{i-1}}^{t_i} b^*(\boldsymbol{x}_t, \boldsymbol{\pi}_{\mathcal{U}}(\boldsymbol{x}_{t_{i-1}}, t_{i-1}, i-1))dt \right], \tag{3}$$

$$t_i = \pi_{\mathcal{T}}(\boldsymbol{x}_{t_{i-1}}, t_{i-1}, i-1) + t_{i-1}, \ t_0 = 0, t_K = T, \forall(\boldsymbol{x}, t, i) : \pi_{\mathcal{T}}(\boldsymbol{x}, t, i) \in [t_{\min}, t_{\max}].$$

In the absence of the transition costs or the bound on the number of interactions, intuitively the policy would propose to interact with the system as frequently as possible, i.e., every $t_{\min}$ seconds. The additional costs/constraints ensure that we do not converge to this trivial (but unrealistic) solution.

## 3 TACOS: Time Adaptive Control or Sensing

In the following, we reformulate the continuous-time problem as an equivalent discrete-time MDP. We first denote the state and running reward flows of Equation (1). The state flow by applying action $\boldsymbol{u}_k$ for $t_k$ time reads:

$$\boldsymbol{x}_{k+1} = \boldsymbol{\Xi}(\boldsymbol{x}_k, \boldsymbol{u}_k, t_k),$$

$$\boldsymbol{\Xi}(\boldsymbol{x}, \boldsymbol{u}, t) \overset{\text{def}}{=} \boldsymbol{x} + \int_0^t \boldsymbol{f}^*(\boldsymbol{x}_s, \boldsymbol{u})ds + \int_0^t \boldsymbol{g}^*(\boldsymbol{x}_s, \boldsymbol{u})d\boldsymbol{B}_s.$$

We assume that every time we interact with the system, we also obtain the integrated reward and define the reward flow as

$$\Xi_{b^*}(\boldsymbol{x}, \boldsymbol{u}, t) \overset{\text{def}}{=} \int_0^t b^*\left(\boldsymbol{\Xi}(\boldsymbol{x}, \boldsymbol{u}, s), \boldsymbol{u}\right) ds. \tag{4}$$

Due to the stochasticity of $(\boldsymbol{B}_t)_{t \in \mathcal{T}}$, the state flow $\boldsymbol{\Xi}(\boldsymbol{x}, \boldsymbol{u}, t)$ and the reward flow $\Xi_{b^*}(\boldsymbol{x}, \boldsymbol{u}, t)$ are stochastic. For ease of notation, we denote

$$\boldsymbol{\Phi}_{\boldsymbol{f}^*}(\boldsymbol{x}_k, \boldsymbol{u}_k, t_k) \overset{\text{def}}{=} \mathbb{E}\left[\boldsymbol{\Xi}(\boldsymbol{x}_k, \boldsymbol{u}_k, t_k)\right], \quad \Phi_{b^*}(\boldsymbol{x}_k, \boldsymbol{u}_k, t_k) \overset{\text{def}}{=} \mathbb{E}\left[\Xi_{b^*}(\boldsymbol{x}_k, \boldsymbol{u}_k, t_k)\right]$$

$$\boldsymbol{w}_k^{\boldsymbol{x}} \overset{\text{def}}{=} \boldsymbol{\Xi}(\boldsymbol{x}_k, \boldsymbol{u}_k, t_k) - \boldsymbol{\Phi}(\boldsymbol{x}_k, \boldsymbol{u}_k, t_k), \quad w_k^{b^*} \overset{\text{def}}{=} \Xi_{b^*}(\boldsymbol{x}_k, \boldsymbol{u}_k, t_k) - \Phi_{b^*}(\boldsymbol{x}_k, \boldsymbol{u}_k, t_k),$$

and the concatenated state and reward flow function, and noise as:

$$\boldsymbol{\Phi}^*(\boldsymbol{x}_k, \boldsymbol{u}_k, t_k) = \begin{pmatrix} \boldsymbol{\Phi}_{\boldsymbol{f}^*}(\boldsymbol{x}_k, \boldsymbol{u}_k, t_k) \\ \Phi_{b^*}(\boldsymbol{x}_k, \boldsymbol{u}_k, t_k) \end{pmatrix}, \quad \boldsymbol{w}_k = \begin{pmatrix} \boldsymbol{w}_k^{\boldsymbol{x}} \\ w_k^{b^*} \end{pmatrix}. \tag{5}$$

In this work, we search for policies that return the next control we apply and also the time for how long to apply the control.

### 3.1 Reforumlation of Interaction Cost setting to Discrete-time MDPs

We convert the problem with interaction costs to a standard MDP which any RL algorithm for continuous state-action spaces can solve. To this end, we restrict ourselves to a policy class:

$$\Pi_{IC} = \{\boldsymbol{\pi} : \mathcal{X} \times \mathcal{T} \to \mathcal{U} \times \mathcal{T} \mid \pi_{\mathcal{T}}(\cdot, t) \in [t_{\min}, t_{\max}], \boldsymbol{\pi} \text{ is } L_{\boldsymbol{\pi}} - \text{Lipschitz}\}.$$

For simplicity, we denote by $\pi_{\mathcal{T}}$ the component of the policy that predicts the duration of applying the action and with $\boldsymbol{\pi}_{\mathcal{U}}$ the component that predicts the action value. The policies we consider map state $\boldsymbol{x}$ and time-to-go $t$ to control $\boldsymbol{u}$ and the time $\tau$ for how long we apply the action $\boldsymbol{u}$. We define the augmented state $\boldsymbol{s} = (\boldsymbol{x}, b, t)$, where $\boldsymbol{x}$ is the state, $b$ integrated reward and $t$ time-to-go. With the introduced notation we arrive at a discrete-time MDP problem formulation

$$\max_{\boldsymbol{\pi} \in \Pi_{IC}} V_{\boldsymbol{\pi}, \boldsymbol{\Phi}^*}(\boldsymbol{x}_0, T) = \max_{\boldsymbol{\pi} \in \Pi_{IC}} \mathbb{E}\left[\sum_{k=0}^{K-1} r(\boldsymbol{s}_k, \boldsymbol{\pi}(\boldsymbol{s}_k))\right] \tag{6}$$

$$\text{s.t.} \quad \boldsymbol{s}_{k+1} = \boldsymbol{\Psi}_{\boldsymbol{\Phi}^*}(\boldsymbol{s}_k, \boldsymbol{\pi}(\boldsymbol{s}_k), \boldsymbol{w}_k), \ \boldsymbol{s}_0 = (\boldsymbol{x}_0, 0, T), \quad \sum_{k=0}^{K-1} \pi_{\mathcal{T}}(\boldsymbol{x}_k, t_k) = T,$$

where we have

$$\boldsymbol{\Psi}_{\boldsymbol{\Phi}^*}(\boldsymbol{s}_k, \boldsymbol{\pi}(\boldsymbol{s}_k), \boldsymbol{w}_k) = (\boldsymbol{\Phi}^*(\boldsymbol{x}_k, \boldsymbol{\pi}(\boldsymbol{x}_k, t_k)) + \boldsymbol{w}_k, t_k - \pi_{\mathcal{T}}(\boldsymbol{x}_k, t_k))$$

$$r(\boldsymbol{s}_k, \boldsymbol{\pi}(\boldsymbol{s}_k)) = \Xi_{b^*}(\boldsymbol{x}_k, \boldsymbol{\pi}(\boldsymbol{x}_k, t_k)) - c(\boldsymbol{x}_k, \boldsymbol{\pi}_{\mathcal{U}}(\boldsymbol{x}_k, t_k)).$$

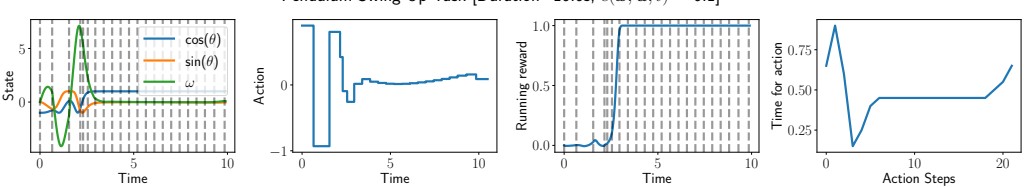

(a) We add a constant switch cost of 0.1 and significantly reduce the number of interactions from 200 to 24. Initially, the policy applies maximal bang-bang torque for longer times, until the pendulum reaches the top. On the top, we measure and change the controller at a higher frequency in order to keep the pendulum stable, at the position with the highest reward.

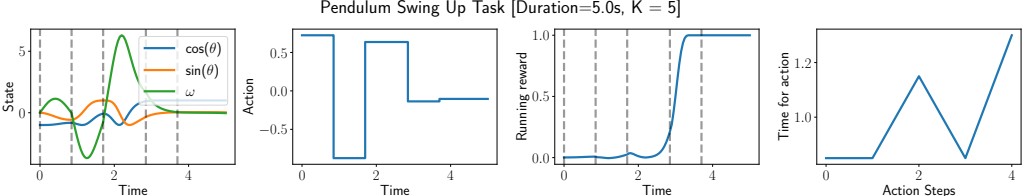

(b) We set a tight bound of $K = 5$ for the number of interactions and observe that we can still solve the task.

Figure 1: Experiment on the Pendulum environment for the average cost and a bounded number of switches setting.

## 3.2 Reformulation of Bounded Number of Interactions to Discrete-time MDPs

The second setting is similar to the one studied by Ni and Jang (2022). In this case, we consider the following class of policies:

$$\Pi_{BI} = \{\boldsymbol{\pi} : \mathcal{X} \times \mathcal{T} \times \mathbb{N} \to \mathcal{U} \times \mathcal{T} \mid \forall k \in [K] : \boldsymbol{\pi}(\cdot, \cdot, k) \text{ is } L_{\boldsymbol{\pi}} - \text{Lipschitz}\}.$$

For an augmented state $\boldsymbol{s} = (\boldsymbol{x}, b, t, k)$, our policies map states $\boldsymbol{x}$, time-to-go $t$, number of past interactions $k$ to a controller $\boldsymbol{u}$ and the time duration $\tau$ for applying the action. Here the optimal control problem reads

$$\max_{\boldsymbol{\pi} \in \Pi_{BI}} V_{\boldsymbol{\pi}, \boldsymbol{\Phi}^*}(\boldsymbol{x}_0, T) = \max_{\boldsymbol{\pi} \in \Pi_{BI}} \mathbb{E}\left[\sum_{k=0}^{K-1} r(\boldsymbol{s}_k, \boldsymbol{\pi}(\boldsymbol{s}_k))\right] \quad (7)$$

$$\text{s.t.} \quad \boldsymbol{s}_{k+1} = \boldsymbol{\Psi}_{\boldsymbol{\Phi}^*}(\boldsymbol{s}_k, \boldsymbol{\pi}(\boldsymbol{s}_k), \boldsymbol{w}_k), \ \boldsymbol{s}_0 = (\boldsymbol{x}_0, 0, T, 0),$$

where,

$$\boldsymbol{\Psi}_{\boldsymbol{\Phi}^*}(\boldsymbol{s}_k, \boldsymbol{\pi}(\boldsymbol{s}_k), \boldsymbol{w}_k) = (\boldsymbol{\Phi}^*(\boldsymbol{x}_k, \boldsymbol{\pi}(\boldsymbol{x}_k, t_k, k)) + \boldsymbol{w}_k, t_k - \pi_{\mathcal{T}}(\boldsymbol{x}_k, t_k, k), k + 1)$$
$$r(\boldsymbol{s}_k, \boldsymbol{\pi}(\boldsymbol{s}_k)) = \Xi_{b^*}(\boldsymbol{x}_k, \boldsymbol{\pi}(\boldsymbol{x}_k, t_k, k)).$$

In the following, we provide a simple proposition which shows that our reformulated problem is equivalent to its continuous-time counterpart from Section 2.

**Proposition 1.** *The problem in Equation* (2) *and 3 are equivalent to Equation* (6) *and 7, respectively.*

Figure 1 depicts the influence of interaction cost and $K$ on the controller's performance for the pendulum environment.

## 4 TACOS with Model-free RL Algorithms

We now illustrate the performance of TACOS on several well-studied robotic RL tasks. We consider the RC car (Kabzan et al., 2020), Greenhouse (Tap, 2000), Pendulum, Reacher, Halfcheetah and Humanoid environments from Brax (Freeman et al., 2021). Thus our experiments range from environments necessitating time-adaptive control like the Greenhouse, a realistic and highly dynamic race car simulation, and a very high dimensional RL task like the Humanoid.[3]

---

[3] $\mathcal{X} \subset \mathbb{R}^{244}, \mathcal{U} \subset \mathbb{R}^{17}$. We provide our implementation at `https://github.com/lasgroup/TaCoS`.

We investigate both the bounded number of interactions and interaction cost settings in our experiments. In particular, we study how the bound $K$ affects the performance of TACoS and compare it to the standard equidistant baseline. We further study the interplay between the stochasticity of the environments (magnitude of $\boldsymbol{g}^*$) and interaction costs and the influence of $t_{\min}$ on TACoS. For all experiments in this section, we combine SAC with TACoS (SAC-TACoS).

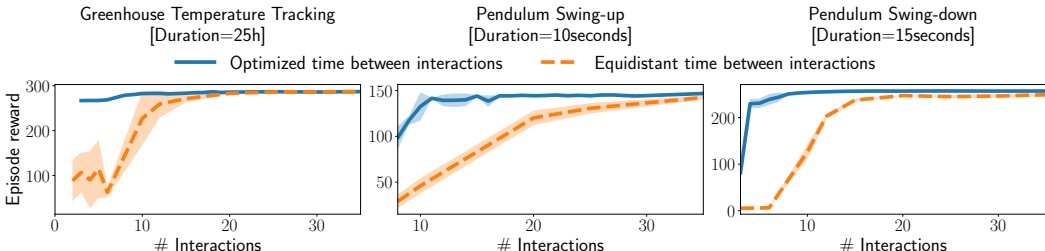

Figure 2: We study the effects of the bound on interactions $K$ on the performance of the agent. TACoS performs significantly better than equidistant discretization, especially for small values of $K$.

**How does the bound on the number of interactions $K$ affect TACoS?** We analyze the bounded number of interactions setting (cf. Section 3.2) of TACoS, studying the relationship between the number of interactions and the achieved episode reward. We compare our algorithm with the standard equidistant time discretization approach which splits the whole horizon $T$ into $T/K$ discrete time steps at which an interaction takes place. We evaluate the two methods in the greenhouse and pendulum environments. For the pendulum, we consider the swing-up and swing-down tasks. The results are reported in Figure 2. The time-adaptive approach performs significantly better than the standard equidistant time discretization. This is particularly the case for the greenhouse and pendulum swing-down tasks. Both tasks involve driving the system to a stable equilibrium and thus, while high-frequency interaction might be necessary at the initial stages, a fairly low interaction frequency can be maintained when the system has reached the equilibrium state. This demonstrates the practical benefits of time-adaptive control.

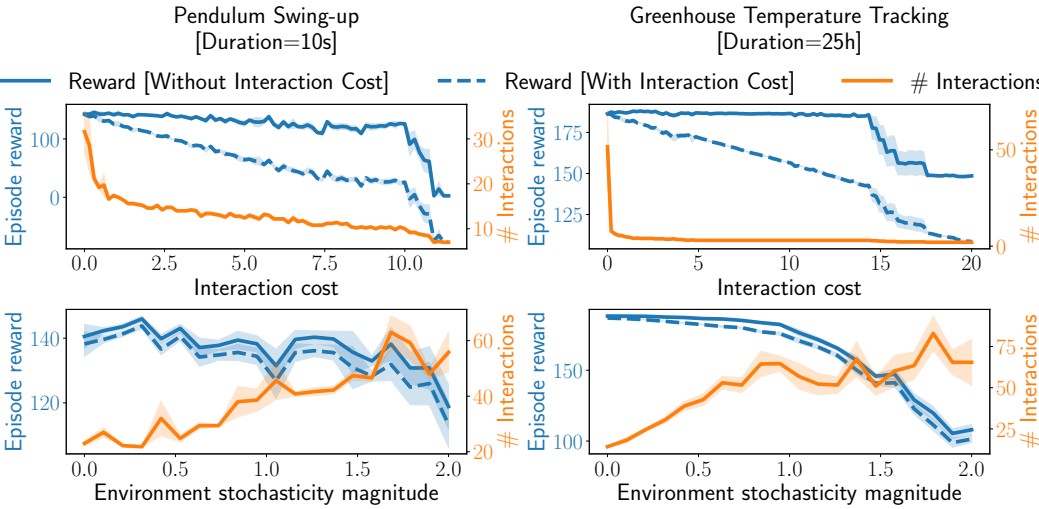

Figure 3: Effect of interaction cost (first row) and environment stochasticity (second row) on the number of interactions and episode reward for the Pendulum and Greenhouse tasks.

**How does the interaction cost magnitude influence TACoS?** We investigate the setting from Section 3.1 with interaction costs. In our experiments, we always pick a constant cost, i.e., $c(\boldsymbol{x}, \boldsymbol{u}) = C$. We study the influence of $C$ on the episode reward and on the number of interactions that the policy has with the system within an episode. We again evaluate this on the greenhouse and pendulum environment. For the pendulum, we consider the swing-up task. The results are presented in the first row of Figure 3. Noticeably, increasing $C$ reduces the number of interactions. The decrease is

drastic for the greenhouse environment since it can be controlled with considerably fewer interactions without having any effect on the performance. Generally, we observe that decreasing the number of interactions, that is, increasing $C$, also results in a slight decline in episode reward.

**How does environment stochasticity influence the number of interactions?** We analyze the influence of the environment's stochasticity, i.e., the magnitude of the diffusion term $g^*$, on the episode reward and number of interactions on TACOS. Intuitively, the more stochastic the environment, the more interactions we would require to stabilize the system. We again evaluate our method on the greenhouse and pendulum swing-up tasks. The results are reported in the second row of Figure 3. The results verify our intuition that more stochasticity in the environment generally leads to more interactions. However, we observe that the policy is still able to achieve high rewards for a wide range of magnitude of $g^*$. This showcases the robustness and adaptability of TACOS to stochastic environments.

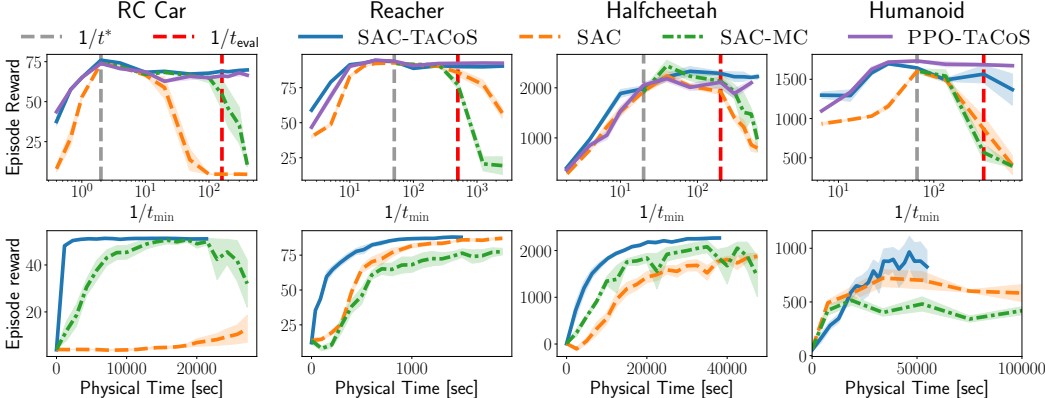

Figure 4: We compare the performance of TACOS in combination with SAC and PPO with the standard SAC algorithm and SAC with more compute (SAC-MC) over a range of values for $t_{\min}$ (first row). In the second row, we plot the episode reward versus the physical time in seconds spent in the environment for SAC-TACOS, SAC, and SAC-MC for a specific evaluation frequency $1/t_{\text{eval}}$. We exclude PPO-TACOS in this plot as it, being on-policy, requires significantly more samples than the off-policy methods. While all methods perform equally well for standard discretization (denoted with $1/t^*$), our method is robust to interaction frequency and does not suffer a performance drop when we decrease $t_{min}$.

**How does $t_{\min}$ influence TACOS?** As highlighted in Section 1, picking the right discretization for interactions is a challenging task. We show that TACOS can naturally alleviate this issue and adaptively pick the frequency of interaction while also being more computationally and data-efficient. Moreover, we show that TACOS is robust to the choice of $t_{\min}$, which represents the minimal duration an action has to be applied, i.e., its inverse is the highest frequency at which we can control the system. In this experiment, we consider SAC-TACOS and compare it to the standard SAC algorithm. TACOS adaptively picks the number of interactions and therefore during an episode of time $T$, it effectively collects less data than the standard discrete-time RL algorithm.[4] This makes comparison to the discrete-time setting challenging since environment interactions and physical time on the environment are not linearly related for TACOS as opposed to the standard discrete-time setting. Nevertheless, to be fair to the discrete-time method, we give SAC more physical time on the system for all environments, effectively resulting in the collection of more data for learning. Since the standard SAC algorithm updates the policy relative to the data amount, we consider a version of SAC, SAC-MC (SAC more compute), which leverages the additional data it collects to perform more gradient updates. This version essentially performs more policy updates than SAC-TACOS and thus is computationally more expensive. Furthermore, to demonstrate the generality of our framework, we also combine TACOS with PPO (PPO-TACOS).

We report the performance after convergence across different $t_{\min}$ in the first row of Figure 4. From our experiment, we conclude that SAC-TACOS and PPO-TACOS are robust to the choice of $t_{\min}$

---

[4]A standard RL algorithm would collect $T/t_{\min}$ data points per episode.

and perform equally well when $t_{\min}$ is decreased, i.e., frequency is increased. This is in contrast to the standard RL methods, which have a significant drop in performance at high frequencies. This observation is also made in prior work (Hafner et al., 2019). Crucially, this highlights the sensitivity of the standard RL methods to the frequency of interaction. In the second row of Figure 4 we show the learning curve of the methods for a specific frequency $1/t_{\text{eval}}$. From the curve, we conclude that SAC-TACOS achieves higher rewards with significantly less physical time on the environment. We believe this is because our method explores more efficiently (akin to Dabney et al., 2020; Eberhard et al., 2022), and also learns a much stronger/continuous-time representation of the underlying MDP.

Interestingly, at the default frequency used in the benchmarks $1/t^*$, all methods perform similarly. However, slightly decreasing the frequency already leads to a drastic drop in performance for all methods. Intuitively, decreasing the frequency prevents us from performing the necessary fine-grained control and obtaining the highest performance.

While we have access to the optimal frequency $1/t^*$ for these benchmarks, for a general and unknown system it is very difficult to estimate this frequency. Furthermore, as we observe in our experiments, picking a very high frequency is also not an option when using standard RL algorithms. We believe this is where TACOS excels as it adaptively picks the frequency of interaction, thereby relieving the problem designer of this decision.

# 5 Efficient Exploration for TACOS via Model-Based RL

In this section, we propose a novel model-based RL algorithm for TACOS called **O**ptimistic **TACOS** (OTACOS). We analyze the episodic setting, where we interact with the system in episodes $n = 1, \ldots, N$. In episode $n$, we execute the policy $\boldsymbol{\pi}_n$, collect measurements and integrated rewards $(\boldsymbol{x}_{n,0}, b_{n,0}), \ldots, (\boldsymbol{x}_{n,k_n}, b_{n,k_n})$, and prepare the data $\mathcal{D}_n = \{(\boldsymbol{z}_{n,1}, \boldsymbol{y}_{n,1}), \ldots, (\boldsymbol{z}_{n,k_n}, \boldsymbol{y}_{n,k_n})\}$, where $\boldsymbol{z}_{n,i} = (\boldsymbol{x}_{n,i-1}, \boldsymbol{u}_{n,i-1}, t_{n,i-1})$ and $\boldsymbol{y}_{n,i} = (\boldsymbol{x}_{n,i}, b_{n,i})$. From the dataset $\mathcal{D}_{1:n} \stackrel{\text{def}}{=} \cup_{i \leq n} \mathcal{D}_i$ we build a model $\mathcal{M}_n$ for the unknown function $\boldsymbol{\Phi}^*$ such that it is well-calibrated in the sense of the following definition.

**Definition 1** (Well-calibrated statistical model of $\boldsymbol{\Phi}^*$, Rothfuss et al. (2023)). *Let $\mathcal{Z} \stackrel{\text{def}}{=} \mathcal{X} \times \mathcal{U} \times \mathcal{T}$. We assume $\boldsymbol{\Phi}^* \in \bigcap_{n \geq 0} \mathcal{M}_n$ with probability at least $1 - \delta$, where statistical model $\mathcal{M}_n$ is defined as*

$$\mathcal{M}_n \stackrel{\text{def}}{=} \left\{ \boldsymbol{f} : \mathcal{Z} \to \mathbb{R}^{d_x+1} \mid \forall \boldsymbol{z} \in \mathcal{Z}, \forall j \in \{1, \ldots, d_x + 1\} : |\mu_{n,j}(\boldsymbol{z}) - f_j(\boldsymbol{z})| \leq \beta_n(\delta)\sigma_{n,j}(\boldsymbol{z}) \right\},$$

*Here, $\mu_{n,j}$ and $\sigma_{n,j}$ denote the $j$-th element in the vector-valued mean and standard deviation functions $\boldsymbol{\mu}_n$ and $\boldsymbol{\sigma}_n$ respectively, and $\beta_n(\delta) \in \mathbb{R}_{\geq 0}$ is a scalar function that depends on the confidence level $\delta \in (0, 1]$ and which is monotonically increasing in $n$.*

Similar to model-based RL algorithms for the discrete-time setting (Kakade et al., 2020; Curi et al., 2020; Sukhija et al., 2024), we follow the principle of optimism in the face of uncertainty and select the policy $\boldsymbol{\pi}_n$ for both settings of TACOS (cf. Sections 3.1 and 3.2) by solving:

$$\boldsymbol{\pi}_n \stackrel{\text{def}}{=} \operatorname*{argmax}_{\boldsymbol{\pi} \in \Pi_\square} \max_{\boldsymbol{\Phi} \in \mathcal{M}_{n-1}} V_{\boldsymbol{\pi}, \boldsymbol{\Phi}}(\boldsymbol{x}_0, T), \tag{8}$$

where $\square \in \{IC, BI\}$ is the appropriate policy class from Section 3. Running OTACOS for $N$ episodes, we measure the performance via the *regret*:

$$R_N = \sum_{n=1}^{N} \left( V_{\boldsymbol{\pi}^*, \boldsymbol{\Phi}^*}(\boldsymbol{x}_0, T) - V_{\boldsymbol{\pi}_n, \boldsymbol{\Phi}^*}(\boldsymbol{x}_0, T) \right).$$

Here $\boldsymbol{\pi}^*$ is the optimal policy from the class of policies we optimize over. Any kind of regret bound requires certain assumptions on the regularity of the underlying dynamics (1).

**Assumption 1** (Dynamics model). *Given any norm $\|\cdot\|$, we assume that the drift $\boldsymbol{f}^*$, and diffusion $\boldsymbol{g}^*$ are $L_{\boldsymbol{f}^*}$ and $L_{\boldsymbol{g}^*}$-Lipschitz continuous, respectively, with respect to the induced metric. We further assume $\sup_{\boldsymbol{z} \in \mathcal{Z}} \|\boldsymbol{g}^*(\boldsymbol{z})\|_F \leq A$.*

Assumption 1 ensures the existence of the SDE (1) solution under policy $\boldsymbol{\pi}_n$. To provide bounds on the performance of OTACOS for settings Sections 3.1 and 3.2 we also need some assumptions on the noise and reward model.

**Assumption 2** (Reward and noise model for Section 3.1 Setting)**.** *Given any norm $\|\cdot\|$, we assume that running reward $b$ is $L_b$-Lipschitz continuous, with respect to the induced metric. We further assume boundedness of the reward $0 \leq b^*(\boldsymbol{x}, \boldsymbol{u}) \leq B$, and interaction cost $0 \leq c(\boldsymbol{x}, \boldsymbol{u}) \leq C$. The dynamics noise is independent and follows: $\boldsymbol{w}_k^{\boldsymbol{x}} \sim \mathcal{N}\left(0, \sigma^2(\boldsymbol{x}_k, \boldsymbol{u}_k, t_k)I_{d_x}\right)$.*

**Assumption 3** (Reward and noise model for Section 3.2 Setting)**.** *Given any norm $\|\cdot\|$, we assume that the running reward $b$ is $L_b$-Lipschitz continuous, w.r.t. to the induced metric.*

Finally, we assume that we learn a well-calibrated model of the unknown flow $\boldsymbol{\Phi}^*$.

**Assumption 4** (Well calibration assumption)**.** *Our learned model is an all-time-calibrated statistical model of $\boldsymbol{\Phi}^*$, i.e., there exists an increasing sequence of $(\beta_n(\delta))_{n \geq 0}$ such that our model satisfies the well-calibration condition, cf., Definition 1.*

Analogous assumptions are made for model-based RL algorithms in the discrete-time setting (Curi et al., 2020; Sukhija et al., 2024). This calibration assumption is satisfied if $\boldsymbol{\Phi}^*$ can be represented with Gaussian Process (GP) (Williams and Rasmussen, 2006; Kirschner and Krause, 2018) models.

**Theorem 2.** *Consider the setting from Section 3.1 and let Assumption 1, 2, and Assumption 4 hold. Then we have with probability at least $1 - \delta$:*

$$R_N \leq \mathcal{O}\left(\beta_{N-1}T^{3/2}\sqrt{N\mathcal{I}_N}\right)$$

*Now consider, the setting with a bounded number of switches $K$, and let Assumption 1, 3, and Assumption 4 hold. Then, we get with probability at least $1 - \delta$*

$$R_N \leq \mathcal{O}\left(\beta_{N-1}^K K e^{D(L_{\boldsymbol{f}^*} + L_{\boldsymbol{g}^*}^2)(1+L_{\boldsymbol{\pi}})TK}\sqrt{N\mathcal{I}_N}\right),$$

*where $D$ is a constant. Here, with $\mathcal{I}_N$ we denote the model-complexity after observing $N$ points (Curi et al., 2020), which quantifies the difficulty of learning $\boldsymbol{\Phi}^*$. For GPs, it behaves similar to the* maximum information gain $\gamma_N$ *(Srinivas et al., 2009), i.e., implying sublinear regret for several common kernels (Vakili et al., 2021).*

As a proof of concept, we evaluate OTACoS on the pendulum and RC car environment for the interaction cost setting. [5] As baselines, we adapt common model-based RL methods such as PETS (Chua et al., 2018) and planning with the mean to TACoS. We call them PETS-TACoS and MEAN-TACoS, respectively. The result is reported in Figure 5. From the figure, we conclude that OTACoS is more sample efficient than other model-based baselines and SAC-TACoS (SAC-TACoS requires circa 6000 episodes for the pendulum and 2000 for the RC car).

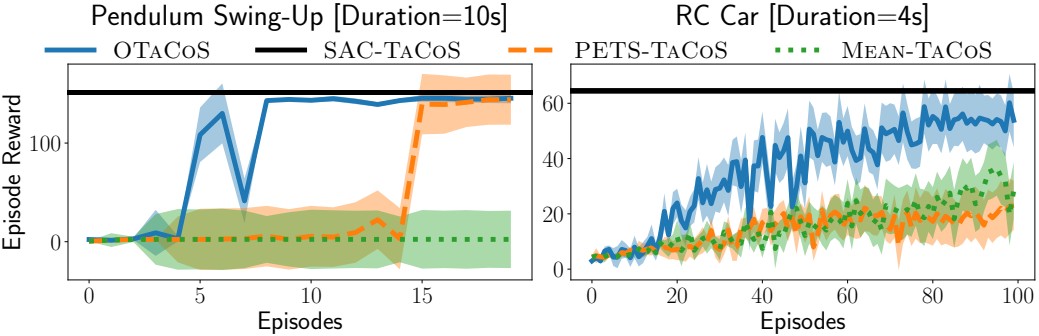

Figure 5: We run OTACoS on the pendulum and RC car environment. We report the achieved reward averaged over five different seeds with one standard error.

## 6 Related Work

Similar to this work, Holt et al. (2023); Ni and Jang (2022); Karimi (2023) consider continuous-time deterministic dynamical systems where the measurements or control input changes can only

---

[5]The code is available at `https://github.com/lasgroup/model-based-rl`.

happen at discrete time steps. Moreover, Holt et al. (2023) proposes a similar problem as ours from Section 3.1, where they specify a cost on the number of interactions. However, their solution is based on a heuristic, where a measurement is taken when the variance of the potential reward surpasses a prespecified threshold. On the contrary, we directly tackle this problem at hand and propose a general framework for time-adaptive control that does not rely on any heuristics. Karimi (2023) adapt SAC (Haarnoja et al., 2018) to include a regularization term, which effectively adds a cost for every discrete interaction. Ni and Jang (2022) induce a soft-constraint on the duration $\tau$ of each action in the environment. However, all the aforementioned works propose heuristic techniques to minimize interactions, whereas we formalize the problem systematically for the more general case of SDEs and show that it has an underlying MDP structure that any RL algorithm can leverage. In addition, we propose a no-regret model-based RL algorithm for this setting and analyze its sample complexity.

Temporal abstractions are considered also in the framework of options (Sutton et al., 1999; Mankowitz et al., 2014; Mann and Mannor, 2014; Harb et al., 2018). However, a key difference to TACoS is that in the options framework, the agent measures the state even between the controller switches.

**Learning to repeat actions** Several works observe that repeating actions in the discrete-time MDPs problems such as Atari (Mnih et al., 2013; Braylan et al., 2015) or Cartpole (Hafner et al., 2019) significantly increase the speed of learning. However, the action repeat is fixed through the entire rollout and treated as a hyperparameter. Durugkar et al. (2016); Vezhnevets et al. (2016); Srinivas et al. (2017); Sharma et al. (2017); Lee et al. (2020); Grigsby et al. (2021); Chen et al. (2021); Nam et al. (2021); Yu et al. (2021); Biedenkapp et al. (2021); Krale et al. (2023) automate the selection of action repeat, and show superior performance over the fixed number setting. Dabney et al. (2020) empirically show that repeating the actions helps with the exploration, effectively having a similar effect that colored noise exploration has over the standard white noise exploration (Eberhard et al., 2022).

**Continuous-time RL** Following the seminal work of Doya (2000) and the advances in Neural ODEs of Chen et al. (2018), continuous-time RL has regained interest (Cranmer et al., 2020; Greydanus et al., 2019; Yildiz et al., 2021; Lutter et al., 2021). Moreover, modeling in continuous-time is found to be particularly useful when learning from different data sources where each source is collected at a different frequency (Burns et al., 2023; Zheng et al., 2023). An important line of work exists for modeling continuous dynamics for the case when states and actions are discrete, called Markov Jump Processes (Kallianpur and Sundar, 2014; Berger, 1993; Huang et al., 2019; Seifner and Sanchez, 2023). Another line of work that is close to ours is event and self-Triggered Control (Astrom and Bernhardsson, 2002; Anta and Tabuada, 2010; Heemels et al., 2012, 2021), where they model continuous-time control systems by implementing changes to the input only when stability is at risk, ensuring efficient and timely interventions. Treven et al. (2023) propose a no-regret continuous-time model-based RL algorithm, which akin to OTACoS, performs optimistic exploration. They study the problem where controls can be executed continuously in time and propose adaptive measurement selection strategies. Similarly, we propose a novel model-based RL algorithm, OTACoS, based on the principle of optimism in the face of uncertainty. We show that OTACoS has no regret for sufficiently smooth dynamics and has considerable sample-efficiency gains over its model-free counterpart.

# 7   Conclusion and discussion

We study the problem of time-adaptive RL for continuous-time systems with continuous state and action spaces. We investigate two practical settings where each interaction has an inherent cost and where we have a hard constraint on the number of interactions. We propose a novel RL framework, TACoS, and show that both of these settings result in extended MDPs which can be solved with standard RL algorithms. In our experiments, we show that combining standard RL algorithms with TACoS results in a significant reduction in the number of interactions without having any effect on the performance for the interaction cost setting. Furthermore, for the second setting, TACoS achieves considerably better control performance despite having a small budget for the number of interactions. Moreover, we show that TACoS improves robustness to a large range of interaction frequencies, and generally improves sample complexity of learning. Finally, we propose, OTACoS, a no-regret model-based RL algorithm for TACoS and show that it has further sample efficiency gains.

## Acknowledgments and Disclosure of Funding

This project has received funding from the Swiss National Science Foundation under NCCR Automation, grant agreement 51NF40 180545, the Microsoft Swiss Joint Research Center, grant of the Hasler foundation (grant no. 21039) and the SNSF Postdoc Mobility Fellowship 211086.

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

# Contents of Appendix

# A Extended Theory

In this section, we prove Theorem 2 for OTACoS. We separate the section into two parts; proof for the transaction cost setting (Appendix A.1) and the proof for the bounded number of switches setting (Appendix A.2).

We start with the definitions of model complexity and sub-Gaussian random vector that we will use extensively in this section.

**Definition 2** (Model Complexity). *We define the model complexity as is defined by Curi et al. (2020).*

$$\mathcal{I}_N := \max_{\mathcal{D}_1,\ldots,\mathcal{D}_N} \sum_{n=1}^{N} \sum_{(\boldsymbol{x},\boldsymbol{u},t)\in\mathcal{D}_n} \|\boldsymbol{\sigma}_n(\boldsymbol{x},\boldsymbol{u},t)\|_2^2. \tag{9}$$

**Definition 3.** *A random variable $x \in \mathbb{R}$ is said to be sub-Gaussian with variance proxy $\sigma^2$ if $\mathbb{E}[x] = 0$ and we have:*

$$\mathbb{E}[e^{tx}] \leq e^{\frac{\sigma^2 t^2}{2}}, \quad \forall t \in \mathbb{R}$$

*A random vector $\boldsymbol{x} \in \mathbb{R}^d$ is said to be sub Gaussian with variance proxy $\sigma^2$ if for any $\boldsymbol{e} \in \mathbb{R}^d$, $\|\boldsymbol{e}\|_2 = 1$ the random variable $\boldsymbol{x}^\top \boldsymbol{e}$ is $\sigma^2$ sub Gaussian. We write $\boldsymbol{x} \sim subG\left(\sigma^2\right)$.*

In the following, we will be distinguishing between the state of the augmented MDP $\boldsymbol{s}$ and the true state of the dynamical system $\boldsymbol{x}$. The augmented state at time step $k$ includes the true state of the system, $\boldsymbol{x}_k$, the integrated reward $b_k$ between $k-1$ and $k$, and the time to left to go $t_k$, i.e., $\boldsymbol{s}_k = [\boldsymbol{x}_k^\top, b_k, t_k]^\top$.

## A.1 Transition Cost setting

We prove our regret bound for the transition cost case in the following. We start with the difference lemma which adapts Sukhija et al. (2024, Lemma 2) to our setting.

**Lemma 3** (Difference lemma). *Define $V_{\boldsymbol{\pi}_n,\boldsymbol{\Phi}}(\boldsymbol{x},\tau)$ as*

$$\mathbb{E}_{\boldsymbol{\pi},\boldsymbol{\Phi}}\left[\sum_{k\geq 0}^{K(\tau)-1} r(\boldsymbol{s}_k,\boldsymbol{\pi}(\boldsymbol{s}_k))\Big|\boldsymbol{x}_0 = \boldsymbol{x}\right] ; \text{ where } \sum_{k=0}^{K(\tau)-1} \pi_{\mathcal{T}}(\boldsymbol{x}_k,t_k) = \tau$$

*that is the total reward starting with time to go $\tau$ and state $\boldsymbol{x}$ for the policy $\boldsymbol{\pi}$ and dynamics $\boldsymbol{\Phi}$. Here the expectation w.r.t. $\boldsymbol{\pi}, \boldsymbol{\Phi}$ represents the expectation of the underlying trajectory induced by the policy $\boldsymbol{\pi}$ on the dynamics $\boldsymbol{\Phi}$. Then we have for all $\boldsymbol{\pi}$, $\boldsymbol{\Phi}'$, $\boldsymbol{\Phi}^*$, $\boldsymbol{x}_0$, $T < 0$;*

$$V_{\boldsymbol{\pi},\boldsymbol{\Phi}'}(\boldsymbol{x}_0,T) - V_{\boldsymbol{\pi},\boldsymbol{\Phi}^*}(\boldsymbol{x}_0,T) = \mathbb{E}_{\boldsymbol{\pi},\boldsymbol{\Phi}^*}\left[\sum_{k\geq 0} V_{\boldsymbol{\pi},\boldsymbol{\Phi}'}(\widehat{\boldsymbol{x}}_{k+1},t_{k+1}) - V_{\boldsymbol{\pi},\boldsymbol{\Phi}'}(\boldsymbol{x}_{k+1},t_{k+1})\right], \tag{10}$$

*where $\widehat{\boldsymbol{x}}_{k+1}$ is the state of $\widehat{\boldsymbol{s}}_{k+1} = \boldsymbol{\Psi}_{\boldsymbol{\Phi}'}(\boldsymbol{s}_k,\boldsymbol{\pi}(\boldsymbol{s}_k),\boldsymbol{w}_k)$ and $\boldsymbol{x}_{k+1}$ is the state of $\boldsymbol{s}_{k+1} = \boldsymbol{\Psi}_{\boldsymbol{\Phi}^*}(\boldsymbol{s}_k,\boldsymbol{\pi}(\boldsymbol{s}_k),\boldsymbol{w}_k)$.*

*Proof.*

$$\begin{aligned}
V_{\boldsymbol{\pi},\boldsymbol{\Phi}^*}(\boldsymbol{x}_0,T) &= \mathbb{E}_{\boldsymbol{\pi},\boldsymbol{\Phi}^*}\left[\sum_{k\geq 0} r(\boldsymbol{s}_k,\boldsymbol{\pi}(\boldsymbol{s}_k))\right] \\
&= \mathbb{E}_{\boldsymbol{\pi},\boldsymbol{\Phi}^*}\left[r(\boldsymbol{s}_0,\boldsymbol{\pi}(\boldsymbol{s}_0)) + \sum_{k\geq 1} r(\boldsymbol{s}_k,\boldsymbol{\pi}(\boldsymbol{s}_k))\right] \\
&= \mathbb{E}_{\boldsymbol{\pi},\boldsymbol{\Phi}^*}\left[r(\boldsymbol{s}_k,\boldsymbol{\pi}(\boldsymbol{s}_0)) + V_{\boldsymbol{\pi},\boldsymbol{\Phi}^*}(\boldsymbol{x}_1,t_1)\right] \\
&= \mathbb{E}_{\boldsymbol{\pi},\boldsymbol{\Phi}^*}\left[r(\boldsymbol{s}_k,\boldsymbol{\pi}(\boldsymbol{s}_0)) + V_{\boldsymbol{\pi},\boldsymbol{\Phi}'}(\widehat{\boldsymbol{x}}_1,t_1) - V_{\boldsymbol{\pi},\boldsymbol{\Phi}'}(\boldsymbol{x}_0,T)\right] + \\
&\quad + \mathbb{E}_{\boldsymbol{\pi},\boldsymbol{\Phi}^*}\left[V_{\boldsymbol{\pi},\boldsymbol{\Phi}}(\boldsymbol{x}_0,T) - V_{\boldsymbol{\pi},\boldsymbol{\Phi}'}(\widehat{\boldsymbol{x}}_1,t_1) + V_{\boldsymbol{\pi},\boldsymbol{\Phi}^*}(\boldsymbol{x}_1,t_1)\right] \\
&= V_{\boldsymbol{\pi},\boldsymbol{\Phi}'}(\boldsymbol{x}_0,T) + \mathbb{E}_{\boldsymbol{\pi},\boldsymbol{\Phi}^*}\left[V_{\boldsymbol{\pi},\boldsymbol{\Phi}}(\boldsymbol{x}_1,t_1) - V_{\boldsymbol{\pi},\boldsymbol{\Phi}'}(\widehat{\boldsymbol{x}}_1,t_1)\right]
\end{aligned}$$

$$+ \mathbb{E}_{\boldsymbol{\pi},\boldsymbol{\Phi}^*}\left[V_{\boldsymbol{\pi},\boldsymbol{\Phi}^*}(\boldsymbol{x}_1,t_1) - V_{\boldsymbol{\pi},\boldsymbol{\Phi}'}(\boldsymbol{x}_1,t_1)\right]$$

Hence we have:

$$V_{\boldsymbol{\pi},\boldsymbol{\Phi}^*}(\boldsymbol{x}_0,T) - V_{\boldsymbol{\pi},\boldsymbol{\Phi}'}(\boldsymbol{x}_0,T) =$$
$$= \mathbb{E}_{\boldsymbol{\pi},\boldsymbol{\Phi}^*}\left[V_{\boldsymbol{\pi},\boldsymbol{\Phi}'}(\boldsymbol{x}_1,t_1) - V_{\boldsymbol{\pi},\boldsymbol{\Phi}'}(\widehat{\boldsymbol{x}}_1,t_1)\right] + \mathbb{E}_{\boldsymbol{\pi},\boldsymbol{\Phi}^*}\left[V_{\boldsymbol{\pi},\boldsymbol{\Phi}^*}(\boldsymbol{x}_1,t_1) - V_{\boldsymbol{\pi},\boldsymbol{\Phi}'}(\boldsymbol{x}_1,t_1)\right]$$

By repeating the step inductively the result follows. $\square$

In the following, we leverage the result above to bound the regret of our optimistic planner w.r.t. the difference in value functions.

**Lemma 4** (Per episode regret bound). *Let Assumption 4 hold, then we have with probability at least $1 - \delta$ for all $n \geq 0$.*

$$V_{\boldsymbol{\pi}_n,\boldsymbol{\Phi}^*}(\boldsymbol{x}_0,T) - V_{\boldsymbol{\pi}^*,\boldsymbol{\Phi}^*}(\boldsymbol{x}_0,T) \leq \mathbb{E}_{\boldsymbol{\pi}_n,\boldsymbol{\Phi}^*}\left[\sum_{k\geq 0} V_{\boldsymbol{\pi}_n,\boldsymbol{\Phi}_n}(\widehat{\boldsymbol{x}}_{n,k+1},t_{n,k+1}) - V_{\boldsymbol{\pi}_n,\boldsymbol{\Phi}_n}(\boldsymbol{x}_{n,k+1},t_{n,k+1})\right].$$
(11)

*Proof.* Since we choose the policy optimistically, we get

$$V_{\boldsymbol{\pi}^*,\boldsymbol{\Phi}^*}(\boldsymbol{x}_0,T) - V_{\boldsymbol{\pi}_n,\boldsymbol{\Phi}^*}(\boldsymbol{x}_0,T) \leq V_{\boldsymbol{\pi}_n,\boldsymbol{\Phi}_n}(\boldsymbol{x}_0,T) - V_{\boldsymbol{\pi}_n,\boldsymbol{\Phi}^*}(\boldsymbol{x}_0,T).$$

Applying Lemma 3 the result follows. $\square$

Now we derive an upper and lower bound on our value function.

**Lemma 5** (Objective upper bound). *Let $\boldsymbol{\pi}$ be any policy from the class $\Pi_{TC}$ and consider any $T > 0$, then we have:*

$$-\frac{C}{t_{\min}}T \leq V_{\boldsymbol{\pi},\boldsymbol{\Psi}^*}(\boldsymbol{x}_0,T) \leq BT.$$

*Proof.* Since running reward is bounded $0 \leq b^*(\boldsymbol{x},\boldsymbol{u}) \leq B$, the number of steps $K$ we can do in an episode is bounded with $0 \leq K \leq \frac{T}{t_{\min}}$, and switch cost is bounded $0 \leq c(\boldsymbol{x},\boldsymbol{u}) \leq C$ we have:

$$-\frac{C}{t_{\min}}T \leq V_{\boldsymbol{\pi},\boldsymbol{\Psi}^*}(\boldsymbol{x}_0,T) \leq BT.$$

$\square$

A key lemma we use to bound the difference in value functions is the following from Kakade et al. (2020).

**Lemma 6** (Absolute expectation Difference Under Two Gaussians (Lemma C.2. Kakade et al. (2020))). *Let $\boldsymbol{z}_1 \sim \mathcal{N}(\boldsymbol{\mu}_1,\sigma^2\boldsymbol{I})$ and $\boldsymbol{z}_2 \sim \mathcal{N}(\boldsymbol{\mu}_2,\sigma^2\boldsymbol{I})$, and for any (appropriately measurable) positive function g, it holds that:*

$$\mathbb{E}[g(\boldsymbol{z}_1)] - \mathbb{E}[g(\boldsymbol{z}_2)] \leq \min\left\{\frac{\|\boldsymbol{\mu}_1 - \boldsymbol{\mu}_2\|}{\sigma^2},1\right\}\sqrt{\mathbb{E}[g^2(\boldsymbol{z}_1)]}$$

Furthermore, due to Assumption 4 we can also bound the distance between the next state prediction by the true system $\boldsymbol{\Phi}^*$ and the optimistic system $\boldsymbol{\Phi}_n$.

**Lemma 7.** *Let Assumption 4 hold, then we have the following for all $n \geq 0$.*

$$\|\boldsymbol{x}_{n,k+1} - \widehat{\boldsymbol{x}}_{n,k+1}\| \leq 2\sqrt{d_{\boldsymbol{x}}}\beta_{n-1}\|\boldsymbol{\sigma}_{n-1}(\boldsymbol{x}_{n,k},\boldsymbol{\pi}_n(\boldsymbol{x}_{n,k},t_{n,k}))\|$$

*Proof.*

$$\|\boldsymbol{x}_{n,k+1} - \widehat{\boldsymbol{x}}_{n,k+1}\| = \|\boldsymbol{\Phi}^*(\boldsymbol{x}_k,\boldsymbol{\pi}_n(\boldsymbol{x}_{n,k},t_{n,k})) + \boldsymbol{w}_{n,k} - (\boldsymbol{\Phi}_n(\boldsymbol{x}_{n,k},\boldsymbol{\pi}_n(\boldsymbol{x}_{n,k},t_{n,k}) + \boldsymbol{w}_{n,k})\|$$
$$= \|\boldsymbol{\Phi}^*(\boldsymbol{x}_k,\boldsymbol{\pi}_n(\boldsymbol{x}_{n,k},t_{n,k})) - \boldsymbol{\Phi}_n(\boldsymbol{x}_{n,k},\boldsymbol{\pi}_n(\boldsymbol{x}_{n,k},t_{n,k}))\|$$
$$\leq 2\sqrt{d_{\boldsymbol{x}}}\beta_{n-1}\|\boldsymbol{\sigma}_{n-1}(\boldsymbol{x}_{n,k},\boldsymbol{\pi}_n(\boldsymbol{x}_{n,k},t_{n,k}))\|,$$

where the last inequality follows from the fact that $\boldsymbol{\Phi}^*,\boldsymbol{\Phi}_n \in \mathcal{M}_{n-1}$ $\square$

Next, we relate the regret at each episode to the model epistemic uncertainty using Lemma 3 and Lemma 7.

**Corollary 8.** *Let Assumption 1 – 2 and Assumption 4 hold, then we have for all $n \geq 0$ with probability at least $1 - \delta$.*

$$V_{\boldsymbol{\pi}_n, \boldsymbol{\Phi}^*}(\boldsymbol{x}_0, T) - V_{\boldsymbol{\pi}^*, \boldsymbol{\Phi}^*}(\boldsymbol{x}_0, T) \leq \frac{2\sqrt{d_{\boldsymbol{x}}}\beta_{n-1}T}{\sigma} \left( B + \frac{C}{t_{\min}} \right) \mathbb{E}\left[ \sum_{k \geq 0} \|\boldsymbol{\sigma}_{n-1}(\boldsymbol{x}_{n,k}, \boldsymbol{\pi}_n(\boldsymbol{x}_{n,k}, t_{n,k}))\| \right]$$
(12)

*Proof.* From Lemma 4 we have:

$$V_{\boldsymbol{\pi}_n, \boldsymbol{\Phi}^*}(\boldsymbol{x}_0, T) - V_{\boldsymbol{\pi}^*, \boldsymbol{\Phi}^*}(\boldsymbol{x}_0, T) \leq \mathbb{E}\left[ \sum_{k \geq 0} V_{\boldsymbol{\pi}_n, \boldsymbol{\Phi}_n}(\boldsymbol{x}_{n,k+1}, t_{n,k+1}) - V_{\boldsymbol{\pi}_n, \boldsymbol{\Phi}_n}(\widehat{\boldsymbol{x}}_{n,k+1}, t_{n,k+1}) \right].$$

Lemma 6 can be applied to positive function $g$. We hence make a transformation and apply it to $g(\cdot) = V_{\boldsymbol{\pi}_n, \boldsymbol{\Phi}_n}(\cdot, t_{n,k+1}) + \frac{C}{t_{\min}}T$, which is positive due to Lemma 5. Moreover, $\forall \boldsymbol{x} \in \mathcal{X}$;

$$g(\cdot) = V_{\boldsymbol{\pi}_n, \boldsymbol{\Phi}_n}(\cdot, t_{n,k+1}) + \frac{C}{t_{\min}}T \leq Bt_{n,k+1} + \frac{C}{t_{\min}}T \leq T(B + \frac{C}{t_{\min}}).$$

Applying Lemma 6 we obtain:

$$V_{\boldsymbol{\pi}_n, \boldsymbol{\Phi}_n}(\boldsymbol{x}_{n,k+1}, t_{n,k+1}) - V_{\boldsymbol{\pi}_n, \boldsymbol{\Phi}_n}(\widehat{\boldsymbol{x}}_{n,k+1}, t_{n,k+1}) \leq \frac{T}{\sigma} \left( B + \frac{C}{t_{\min}} \right) \mathbb{E}\left[ \|\boldsymbol{x}_{n,k+1} - \widehat{\boldsymbol{x}}_{n,k+1}\| \right]$$

Finally, applying Lemma 7 we arrive at:

$$V_{\boldsymbol{\pi}_n, \boldsymbol{\Phi}^*}(\boldsymbol{x}_0, T) - V_{\boldsymbol{\pi}^*, \boldsymbol{\Phi}^*}(\boldsymbol{x}_0, T) \leq \frac{2\sqrt{d_{\boldsymbol{x}}}\beta_{n-1}T}{\sigma} \left( B + \frac{C}{t_{\min}} \right) \mathbb{E}\left[ \sum_{k \geq 0} \|\boldsymbol{\sigma}_{n-1}(\boldsymbol{x}_{n,k}, \boldsymbol{\pi}_n(\boldsymbol{x}_{n,k}, t_{n,k}))\| \right]$$

$\square$

Now we can prove our regret bound for the transition cost case.

**Theorem 9.** *Let Assumption 1 – 2 and Assumption 4 hold, then we have for all $n \geq 0$ with probability at least $1 - \delta$.*

$$R_N = \sum_{n=1}^{N} V_{\boldsymbol{\pi}_n, \boldsymbol{\Phi}^*}(\boldsymbol{x}_0, T) - V_{\boldsymbol{\pi}^*, \boldsymbol{\Phi}^*}(\boldsymbol{x}_0, T)$$

$$\leq \frac{2\sqrt{d_{\boldsymbol{x}}}\beta_{N-1}T^{3/2}}{\sigma^2 t_{\min}} \left( B + \frac{C}{t_{\min}} \right) \sqrt{N\mathcal{I}_N}$$

*Proof.* We compute:

$$R_N = \sum_{n=1}^{N} V_{\boldsymbol{\pi}_n, \boldsymbol{\Phi}^*}(\boldsymbol{x}_0, T) - V_{\boldsymbol{\pi}^*, \boldsymbol{\Phi}^*}(\boldsymbol{x}_0, T)$$

$$\leq \frac{2\sqrt{d_{\boldsymbol{x}}}T}{\sigma} \left( B + \frac{C}{t_{\min}} \right) \sum_{n=1}^{N} \beta_{n-1} \mathbb{E}\left[ \sum_{k \geq 0} \|\boldsymbol{\sigma}_{n-1}(\boldsymbol{x}_{n,k}, \boldsymbol{\pi}_n(\boldsymbol{x}_{n,k}, t_{n,k}))\| \right]$$

$$\leq \frac{2\sqrt{d_{\boldsymbol{x}}}\beta_{N-1}T}{\sigma} \left( B + \frac{C}{t_{\min}} \right) \mathbb{E}\left[ \sum_{n=1}^{N} \sum_{k \geq 0} \|\boldsymbol{\sigma}_{n-1}(\boldsymbol{x}_{n,k}, \boldsymbol{\pi}_n(\boldsymbol{x}_{n,k}, t_{n,k}))\| \right]$$

$$\leq \frac{2\sqrt{d_{\boldsymbol{x}}}\beta_{N-1}T}{\sigma} \left( B + \frac{C}{t_{\min}} \right) \sqrt{\frac{TN}{t_{\min}}} \mathbb{E}\left[ \sqrt{\sum_{n=1}^{N} \sum_{k \geq 0} \|\boldsymbol{\sigma}_{n-1}(\boldsymbol{x}_{n,k}, \boldsymbol{\pi}_n(\boldsymbol{x}_{n,k}, t_{n,k}))\|^2} \right]$$

$$\leq \frac{2\sqrt{d_{\boldsymbol{x}}}\beta_{N-1}T^{3/2}}{\sigma\sqrt{t_{\min}}}\left(B+\frac{C}{t_{\min}}\right)\sqrt{N\mathcal{I}_N}$$

Here the first inequality follows because of Corollary 8, the second inequality follows due to the monotonicity of sequence $(\beta_n)_{n\geq 0}$, the third inequality follows by Cauchy–Schwarz and the last one by maximizing the term in expectation. $\qquad\square$

Our regret $R_N$ is sublinear if $\beta_{N-1}\sqrt{N\mathcal{I}_N}$ is sublinear. For general well-calibrated models this is tough to verify. However, for Gaussian process dynamics, $\mathcal{I}_N$ is equal to (up to constant factors) the maximum information gain $\gamma_N$ (Srinivas et al., 2009) (c.f., Curi et al. (2020, Lemma 17)). The maximum information gain is sublinear for a rich class of kernels (Vakili et al., 2021), i.e., yielding sublinear regret for OTACOS (see Sukhija et al. (2024, Theorem 2) for more detail).

### A.2 Bounded number of transition

We overload the notation in this section and add number of switches to the value function, such that we have $V_{\boldsymbol{\pi}_n,\boldsymbol{\Phi}^*}(\boldsymbol{x}_0,T,0)=V_{\boldsymbol{\pi}_n,\boldsymbol{\Phi}^*}(\boldsymbol{x}_0,T)$

**Lemma 10** (Per episode regret bound). *We have:*

$$V_{\boldsymbol{\pi}_n,\boldsymbol{\Phi}^*}(\boldsymbol{x}_0,T,0)-V_{\boldsymbol{\pi}^*,\boldsymbol{\Phi}^*}(\boldsymbol{x}_0,T,0)\leq$$

$$\leq\mathbb{E}\left[\sum_{k=0}^{K-1}V_{\boldsymbol{\pi}_n,\boldsymbol{\Phi}_n}(\boldsymbol{x}_{n,k+1},t_{n,k+1},k+1)-V_{\boldsymbol{\pi}_n,\boldsymbol{\Phi}_n}(\widehat{\boldsymbol{x}}_{n,k+1},t_{n,k+1},k+1)\right],$$

*where $\widehat{\boldsymbol{x}}_{n,k+1}$ is the state of one step hallucinated component $\widehat{\boldsymbol{s}}_{n,k+1}=\boldsymbol{\Psi}_{\boldsymbol{\Phi}_n}(\boldsymbol{s}_{n,k},\boldsymbol{\pi}_n(\boldsymbol{s}_{n,k}),\boldsymbol{w}_{n,k})$ and $\boldsymbol{x}_{n,k+1}$ is the state of $\boldsymbol{s}_{n,k+1}=\boldsymbol{\Psi}_{\boldsymbol{\Phi}^*}(\boldsymbol{s}_{n,k},\boldsymbol{\pi}_n(\boldsymbol{s}_{n,k}),\boldsymbol{w}_{n,k})$.*

*Proof.*

$$\begin{aligned}
V_{\boldsymbol{\pi}_n,\boldsymbol{\Phi}^*}(\boldsymbol{x}_0,T,0)&=\mathbb{E}\left[\sum_{k\geq 0}r(\boldsymbol{s}_{n,k},\boldsymbol{\pi}_n(\boldsymbol{s}_{n,k}))\right]=\mathbb{E}\left[r(\boldsymbol{s}_{n,0},\boldsymbol{\pi}_n(\boldsymbol{s}_{n,0}))+\sum_{k\geq 1}r(\boldsymbol{s}_{n,k},\boldsymbol{\pi}_n(\boldsymbol{s}_{n,k}))\right]\\
&=\mathbb{E}\left[r(\boldsymbol{s}_{n,k},\boldsymbol{\pi}_n(\boldsymbol{s}_{n,0}))+V_{\boldsymbol{\pi}_n,\boldsymbol{\Phi}^*}(\boldsymbol{x}_{n,1},t_{n,1},1)\right]\\
&=\mathbb{E}\left[r(\boldsymbol{s}_{n,k},\boldsymbol{\pi}_n(\boldsymbol{s}_{n,0}))+V_{\boldsymbol{\pi}_n,\boldsymbol{\Phi}_n}(\boldsymbol{x}_{n,1},t_{n,1},1)-V_{\boldsymbol{\pi}_n,\boldsymbol{\Phi}_n}(\boldsymbol{x}_0,T,0)\right]+\\
&\quad+\mathbb{E}\left[V_{\boldsymbol{\pi}_n,\boldsymbol{\Phi}_n}(\boldsymbol{x}_0,T,0)-V_{\boldsymbol{\pi}_n,\boldsymbol{\Phi}_n}(\boldsymbol{x}_{n,1},t_{n,1},1)+V_{\boldsymbol{\pi}_n,\boldsymbol{\Phi}^*}(\boldsymbol{x}_{n,1},t_{n,1},1)\right]\\
&=V_{\boldsymbol{\pi}_n,\boldsymbol{\Phi}_n}(\boldsymbol{x}_0,T,0)+\mathbb{E}\left[V_{\boldsymbol{\pi}_n,\boldsymbol{\Phi}_n}(\widehat{\boldsymbol{x}}_{n,1},t_{n,1},1)-V_{\boldsymbol{\pi}_n,\boldsymbol{\Phi}_n}(\boldsymbol{x}_{n,1},t_{n,1},1)\right]\\
&\quad+\mathbb{E}\left[V_{\boldsymbol{\pi}_n,\boldsymbol{\Phi}^*}(\boldsymbol{x}_{n,1},t_{n,1},1)-V_{\boldsymbol{\pi}_n,\boldsymbol{\Phi}_n}(\boldsymbol{x}_{n,1},t_{n,1},1)\right]
\end{aligned}$$

Hence we have:

$$V_{\boldsymbol{\pi}_n,\boldsymbol{\Phi}^*}(\boldsymbol{x}_0,T,0)-V_{\boldsymbol{\pi}_n,\boldsymbol{\Phi}_n}(\boldsymbol{x}_0,T,0)=$$
$$=\mathbb{E}\left[V_{\boldsymbol{\pi}_n,\boldsymbol{\Phi}_n}(\widehat{\boldsymbol{x}}_{n,1},t_{n,1},1)-V_{\boldsymbol{\pi}_n,\boldsymbol{\Phi}_n}(\boldsymbol{x}_{n,1},t_{n,1},1)\right]+\mathbb{E}\left[V_{\boldsymbol{\pi}_n,\boldsymbol{\Phi}^*}(\boldsymbol{x}_{n,1},t_{n,1},1)-V_{\boldsymbol{\pi}_n,\boldsymbol{\Phi}_n}(\boldsymbol{x}_{n,1},t_{n,1},1)\right]$$

Repeating the step inductively the result follows and using $V_{\boldsymbol{\pi}_n,\boldsymbol{\Phi}^*}(\boldsymbol{x}_{n,K},t_{n,K},K)=0$ we prove the lemma. $\qquad\square$

#### A.2.1 Subgaussianity of the noise

In principle, we could assume that the noise $\boldsymbol{w}_k$ is Gaussian and then with the same analysis obtain the regret bound. However, stochastic flows are in many cases not exactly Gaussian but only sub-Gaussian. For such noise we need can not apply Lemma 6 and need to escort to different analysis. First we show that under mild assumptions on the SDE dynamics functions $\boldsymbol{f}^*$ and $\boldsymbol{g}^*$ the resulting noise $\boldsymbol{w}_k$ is sub-Gaussian.

To derive this result we will follow the work of Djellout et al. (2004). We present the results in quite informal way, for more rigorous statements we refer the reader to Djellout et al. (2004).

**Definition 4** (Wasserstein distance). *Let $(\mathcal{E},d_{\mathcal{E}})$ be a metric space and let $\mu,\nu$ be two probability measures on $\mathcal{E}$. We define:*

$$W_p(\mu,\nu)=\inf_{\gamma\in\Gamma(\mu,\nu)}\mathbb{E}_{(x,y)\sim\gamma}\left[d(x,y)^p\right]^{\frac{1}{p}}$$

**Definition 5** (Kullback–Leibler divergence). *Let $(\mathcal{E}, d_{\mathcal{E}})$ be a metric space and let $\mu, \nu$ be two probability measures on $\mathcal{E}$. We define:*

$$H(\nu||\mu) = \begin{cases} \mathbb{E}_{x \sim \nu} \left[ \log \left( \frac{d\nu(x)}{d\mu(x)} \right) \right], & \text{if } \nu \ll \mu \\ +\infty, & \text{else} \end{cases}$$

**Definition 6** ($L^p$-transportation cost information inequality). *Let $(\mathcal{E}, d_{\mathcal{E}})$ be a metric space and let $\mu$ be a probability measure on $\mathcal{E}$. We say that $\mu$ satisfy the $L^p$-transportation cost information inequality, and for short write $\mu \in T_p(C)$, if there exists a constant $C$ such that for any measure $\nu$ on $\mathcal{E}$ we have:*

$$W_p(\mu, \nu) \leq \sqrt{2CH(\nu||\mu)}.$$

We now state an important theroem of Bobkov and Götze (1999) that we will use later.

**Theorem 11** (From Bobkov and Götze (1999)). *Let $(\mathcal{E}, d_{\mathcal{E}})$ be a metric space and let $\mu$ be a probability measure on $\mathcal{E}$. We have that $\mu \in T_1(C)$ if and only if for any $\mu$-integrable and $L_F$-Lipschitz function $F : (\mathcal{E}, d_{\mathcal{E}}) \to \mathbb{R}$ and for any $\lambda \in \mathbb{R}$ we have:*

$$\mathbb{E}_{x \sim \mu} \left[ e^{\lambda(F(x) - \mathbb{E}_{x \sim \mu}[F(x)])} \right] \leq e^{\frac{\lambda^2}{2} C L_F^2}$$

Next, we provide a condition under which $\boldsymbol{\Xi}(\boldsymbol{x}, \boldsymbol{u}, t)$ is sub-Gaussian random variable for any $t \in \mathcal{T}$.

**Corollary 12** (Adjusted Corollary 4.1 of Djellout et al. (2004)). *Assume*

$$\sup_{\substack{\boldsymbol{x} \in \mathbb{R}^{d_x} \\ \boldsymbol{u} \in \mathbb{R}^{d_u}}} \|\boldsymbol{g}^*(\boldsymbol{x}, \boldsymbol{u})\|_F \leq A, \quad \|\boldsymbol{f}^*(\boldsymbol{x}, \boldsymbol{u}) - \boldsymbol{f}^*(\widehat{\boldsymbol{x}}, \widehat{\boldsymbol{u}})\| \leq L_{\boldsymbol{f}^*} \|(\boldsymbol{x}, \boldsymbol{u}) - (\widehat{\boldsymbol{x}}, \widehat{\boldsymbol{u}})\|,$$

*and denote the law of $(\boldsymbol{\Xi}(\boldsymbol{x}, \boldsymbol{u}, t))_{t \in \mathcal{T}}$ on the space $C(\mathcal{T}, \mathbb{R}^{d_x})$ (space of continuous functions from $\mathcal{T}$ to $\mathbb{R}^{d_x}$) by $\mathbb{P}_{\boldsymbol{x}}$. Then, there exist a constant $C = C(A, L_{\boldsymbol{f}^*}, T)$ such that $\mathbb{P}_{\boldsymbol{x}} \in T_1(C)$ on the space $C(\mathcal{T}, \mathbb{R}^{d_x})$ equipped with the metric:*

$$d(\gamma_1, \gamma_2) = \sup_{t \in [0,T]} \|\gamma_1(t) - \gamma_2(t)\|$$

Lets $\boldsymbol{e}$ be a(ny) unit vector in $\mathbb{R}^{d_x}$ and define:

$$F_{\boldsymbol{e}, t} : C(\mathcal{T}, \mathbb{R}^{d_x}) \to \mathbb{R}$$
$$F_{\boldsymbol{e}, t} : \gamma \mapsto \gamma(t)^\top \boldsymbol{e}$$

We have:

$$\begin{aligned} |F_{\boldsymbol{e}, t}(\gamma_1) - F_{\boldsymbol{e}, t}(\gamma_2)| &= \left| (\gamma_1(t) - \gamma_2(t))^\top \boldsymbol{e} \right| \\ &\leq \|\gamma_1(t) - \gamma_2(t)\| \|e\| = \|\gamma_1(t) - \gamma_2(t)\| \\ &\leq \sup_{t \in \mathcal{T}} \|\gamma_1(t) - \gamma_2(t)\| = d(\gamma_1, \gamma_2) \end{aligned}$$

Therefore for any $\boldsymbol{e}, t$ the function $F_{\boldsymbol{e}, t}$ is 1–Lipschitz. Since we have

$$\mathbb{E}[|F_{\boldsymbol{e}, t}(\gamma)|] = \int_{C(\mathcal{T}, \mathbb{R}^{d_x})} |\gamma(t)| \, d\mathbb{P}_{\boldsymbol{x}}(\gamma) = \mathbb{E}[|\boldsymbol{\Xi}(\boldsymbol{x}, \boldsymbol{u}, t)^\top \boldsymbol{e}|] < \infty$$

the function $F_{\boldsymbol{e}, t}$ is also $\mathbb{P}_{\boldsymbol{x}}$-integrable. Combining the latter observation with the Theorem 11 we obtain that for any $\boldsymbol{e} \in \mathbb{R}^{d_x}$ and any $t \in \mathcal{T}$ we have:

$$\mathbb{E}_{\boldsymbol{\Xi}(\boldsymbol{x}, \boldsymbol{u}, t)} \left[ e^{\lambda(\boldsymbol{\Xi}(\boldsymbol{x}, \boldsymbol{u}, t)^\top \boldsymbol{e} - \mathbb{E}[\boldsymbol{\Xi}(\boldsymbol{x}, \boldsymbol{u}, t)^\top \boldsymbol{e}])} \right] = \mathbb{E}_{\gamma \sim \mathbb{P}_{\boldsymbol{x}}} \left[ e^{\lambda(F_{\boldsymbol{e}, t}(\gamma) - \mathbb{E}_{\gamma \sim \mathbb{P}_{\boldsymbol{x}}}[F_{\boldsymbol{e}, t}(\gamma)])} \right] \leq e^{\frac{\lambda^2}{2} C}$$

Hence under the assumption of Theorem 2 for Bounded number of switches setting we have that for any $t \in \mathcal{T}$ the random variable $\boldsymbol{\Xi}(\boldsymbol{x}, \boldsymbol{u}, t) - \mathbb{E}[\boldsymbol{\Xi}(\boldsymbol{x}, \boldsymbol{u}, t)] \sim \text{subG}(C)$. The variance proxy $C$ depends on $A, L_{\boldsymbol{f}^*}, T$.

### A.2.2 Lipschitness of the expected flow $\Phi^*$

To apply analysis for the case when noise $\boldsymbol{w}_k$ is any sub-Gaussian we also need to show that the dynamics function $\boldsymbol{\Phi}^*$ is Lipschitz. We first start with some general results.

**Lemma 13.** *Let $\boldsymbol{f} : \mathbb{R}^n \to \mathbb{R}^m$, $A \subset [n]$ and denote $B = A^C$. If we have:*

- $\|\boldsymbol{f}(\boldsymbol{x}_A, \boldsymbol{x}_B) - \boldsymbol{f}(\widehat{\boldsymbol{x}}_A, \boldsymbol{x}_B)\|_2 \le L_A \|\boldsymbol{x}_A - \widehat{\boldsymbol{x}}_A\|_2$,

- $\|\boldsymbol{f}(\boldsymbol{x}_A, \boldsymbol{x}_B) - \boldsymbol{f}(\boldsymbol{x}_A, \widehat{\boldsymbol{x}}_B)\|_2 \le L_B \|\boldsymbol{x}_B - \widehat{\boldsymbol{x}}_B\|_2$,

*then $\boldsymbol{f}$ is $2(L_A + L_B)$ Lipschitz.*

*Proof.* We have:

$$
\begin{aligned}
\|\boldsymbol{f}(\boldsymbol{x}) - \boldsymbol{f}(\widehat{\boldsymbol{x}})\|_2 &= \|\boldsymbol{f}(\boldsymbol{x}_A, \boldsymbol{x}_B) - \boldsymbol{f}(\widehat{\boldsymbol{x}}_A, \widehat{\boldsymbol{x}}_B)\|_2 \\
&= \|\boldsymbol{f}(\boldsymbol{x}_A, \boldsymbol{x}_B) - \boldsymbol{f}(\widehat{\boldsymbol{x}}_A, \boldsymbol{x}_B) + \boldsymbol{f}(\widehat{\boldsymbol{x}}_A, \boldsymbol{x}_B) - \boldsymbol{f}(\widehat{\boldsymbol{x}}_A, \widehat{\boldsymbol{x}}_B)\|_2 \\
&\le L_A \|\boldsymbol{x}_A - \widehat{\boldsymbol{x}}_A\|_2 + L_B \|\boldsymbol{x}_B - \widehat{\boldsymbol{x}}_B\|_2 \\
&\le (L_A + L_B)\left(\|\boldsymbol{x}_A - \widehat{\boldsymbol{x}}_A\|_2 + \|\boldsymbol{x}_B - \widehat{\boldsymbol{x}}_B\|_2\right) \\
&\le 2(L_A + L_B)\left\|\begin{pmatrix} \boldsymbol{x}_A - \widehat{\boldsymbol{x}}_A \\ \boldsymbol{x}_B - \widehat{\boldsymbol{x}}_B \end{pmatrix}\right\|_2 \\
&= 2(L_A + L_B)\|\boldsymbol{x} - \widehat{\boldsymbol{x}}\|_2
\end{aligned}
$$

$\square$

**Lemma 14** (Lipschitzness of $\boldsymbol{\Phi}_{\boldsymbol{f}^*}$). *There exists a positive constant $L_{\boldsymbol{\Phi}_{\boldsymbol{f}}}$ such that the flow $\boldsymbol{\Phi}_{\boldsymbol{f}^*}$ is $L_{\boldsymbol{\Phi}_{\boldsymbol{f}}}$–Lipschitz.*

*Proof.* We will first prove coordinate-wise Lipschitzness. We observe:

1. Lipschitness in time:

$$
\begin{aligned}
\left\|\boldsymbol{\Phi}_{\boldsymbol{f}^*}(\boldsymbol{x}, \boldsymbol{u}, t) - \boldsymbol{\Phi}_{\boldsymbol{f}^*}(\boldsymbol{x}, \boldsymbol{u}, \widehat{t})\right\| &= \left\|\int_0^t \mathbb{E}[\boldsymbol{f}^*(\boldsymbol{x}_s, \boldsymbol{u})]ds - \int_0^{\widehat{t}} \mathbb{E}[\boldsymbol{f}^*(\boldsymbol{x}_s, \boldsymbol{u})]ds\right\| \\
&\le \int_{\widehat{t}}^t \mathbb{E}\left[\|\boldsymbol{f}^*(\boldsymbol{x}_s, \boldsymbol{u})\|\right]ds \le F\left|t - \widehat{t}\right|
\end{aligned}
$$

2. Lipschitness in state $\boldsymbol{x}$: To prove this, consider the $\delta\boldsymbol{x}_t = \boldsymbol{\Xi}(\boldsymbol{x}, \boldsymbol{u}, t) - \boldsymbol{\Xi}(\widehat{\boldsymbol{x}}, \boldsymbol{u}, t)$, then we have

$$
\begin{aligned}
d\delta\boldsymbol{x}_t &= (\boldsymbol{f}^*(\boldsymbol{x}_t, \boldsymbol{u}) - \boldsymbol{f}^*(\widehat{\boldsymbol{x}}_t, \boldsymbol{u}))dt + (\boldsymbol{g}^*(\boldsymbol{x}_t, \boldsymbol{u}) - \boldsymbol{f}^*(\widehat{\boldsymbol{x}}_t, \boldsymbol{u}))d\boldsymbol{B}_t \\
&= \delta\boldsymbol{f}_t^* dt + \delta\boldsymbol{g}_t^* d\boldsymbol{B}_t.
\end{aligned}
$$

Note that $\|\delta\boldsymbol{f}_t^*\| \le L_{\boldsymbol{f}^*}\|\delta\boldsymbol{x}_t\|$ and $\|\delta\boldsymbol{g}_t^*\| \le L_{\boldsymbol{g}^*}\|\delta\boldsymbol{x}_t\|$ since both functions are Lipschitz. Define $\boldsymbol{y}_t = \delta\boldsymbol{x}_t^\top \delta\boldsymbol{x}_t$ and use Ito's Lemma to get

$$
d\boldsymbol{y}_t = 2\delta\boldsymbol{x}_t^\top(\delta\boldsymbol{f}_t^* dt + \delta\boldsymbol{g}_t^* d\boldsymbol{B}_t) + \text{tr}(\delta\boldsymbol{g}_t^*(\delta\boldsymbol{g}_t^*)^\top)dt
$$

Moreover,

$$
\begin{aligned}
\mathbb{E}[\boldsymbol{y}_t] &= \int_0^t 2\mathbb{E}[\delta\boldsymbol{x}_s^\top \delta\boldsymbol{f}_s^*] + \mathbb{E}[\text{tr}(\delta\boldsymbol{g}_s^*(\delta\boldsymbol{g}_s^*)^\top)]ds \\
&\le \int_0^t 2\mathbb{E}\left[\|\delta\boldsymbol{x}_s\|\,\|\delta\boldsymbol{f}_s^*\|\right] + \mathbb{E}[\|\delta\boldsymbol{g}_s^*\|^2]ds \\
&\le \int_0^t (2L_{\boldsymbol{f}^*} + L_{\boldsymbol{g}^*}^2)\mathbb{E}[\|\delta\boldsymbol{x}_s\|^2]ds
\end{aligned}
$$

Note that $\boldsymbol{y}_t = \|\delta\boldsymbol{x}_t\|^2$, so we can apply Grönwall's inequality to get

$$
\mathbb{E}\left[\|\delta\boldsymbol{x}_t\|^2\right] \le \|\delta\boldsymbol{x}_0\|^2 e^{(2L_{\boldsymbol{f}^*} + L_{\boldsymbol{g}^*}^2)t}.
$$

Moreover,

$$\|\mathbb{E}[\delta \boldsymbol{x}_t]\| \leq \sqrt{\mathbb{E}\left[\|\delta \boldsymbol{x}_t\|^2\right]} \leq \|\delta \boldsymbol{x}_0\| e^{\frac{2L_{\boldsymbol{f}*}+L_{\boldsymbol{g}*}^2}{2}t} \leq \|\delta \boldsymbol{x}_0\| e^{\frac{2L_{\boldsymbol{f}*}+L_{\boldsymbol{g}*}^2}{2}T}.$$

Hence we have:

$$\|\boldsymbol{\Phi}_{\boldsymbol{f}*}(\boldsymbol{x}, \boldsymbol{u}, t) - \boldsymbol{\Phi}_{\boldsymbol{f}*}(\widehat{\boldsymbol{x}}, \boldsymbol{u}, t)\| \leq \|\boldsymbol{x} - \widehat{\boldsymbol{x}}\| e^{\frac{2L_{\boldsymbol{f}*}+L_{\boldsymbol{g}*}^2}{2}T}.$$

3. Lipschitzness in action $\boldsymbol{u}$: We denote $\delta \boldsymbol{x}_t = \boldsymbol{\Xi}(\boldsymbol{x}, \boldsymbol{u}, t) - \boldsymbol{\Xi}(\boldsymbol{x}, \widehat{\boldsymbol{u}}, t)$ and $\delta \boldsymbol{u} = \boldsymbol{u} - \widehat{\boldsymbol{u}}$ Following the same steps as in the proof of Lipschitzness in state we arrive at:

$$d\boldsymbol{y}_t = 2\delta \boldsymbol{x}_t^\top (\delta \boldsymbol{f}_t^* dt + \delta \boldsymbol{g}_t^* d\boldsymbol{B}_t) + \text{tr}(\delta \boldsymbol{g}_t^* (\delta \boldsymbol{g}_t^*)^\top)dt$$

Integration yields:

$$\mathbb{E}[\boldsymbol{y}_t] = \int_0^t 2\mathbb{E}[\delta \boldsymbol{x}_s^\top \delta \boldsymbol{f}_s^*] + \mathbb{E}[\text{tr}(\delta \boldsymbol{g}_s^* (\delta \boldsymbol{g}_s^*)^\top)]ds$$

$$\leq \int_0^t 2\mathbb{E}\left[\|\delta \boldsymbol{x}_s\| \|\delta \boldsymbol{f}_s^*\|\right] + \mathbb{E}[\|\delta \boldsymbol{g}_s^*\|^2]ds$$

$$\leq \int_0^t 2\mathbb{E}\left[L_{\boldsymbol{f}*} \|\delta \boldsymbol{x}_s\| (\|\delta \boldsymbol{x}_s\| + \|\delta \boldsymbol{u}\|)\right] + \mathbb{E}\left[2L_{\boldsymbol{g}*}^2 \left(\|\delta \boldsymbol{x}_s\|^2 + \|\delta \boldsymbol{u}\|^2\right)\right]ds$$

$$\leq \int_0^t (3L_{\boldsymbol{f}*} + 2L_{\boldsymbol{g}*}^2)\mathbb{E}[\boldsymbol{y}_s] + (L_{\boldsymbol{f}*} + 2L_{\boldsymbol{g}*}^2)\|\delta \boldsymbol{u}\|ds,$$

where we used $(a+b)^2 \leq 2a^2 + 2b^2$ and $ab \leq \frac{1}{2}(a^2 + b^2)$. Applying Grönwall's inequality results in:

$$\mathbb{E}\left[\|\delta \boldsymbol{x}_t\|^2\right] \leq \|\delta \boldsymbol{u}\|^2 (L_{\boldsymbol{f}*} + 2L_{\boldsymbol{g}*}^2)e^{(3L_{\boldsymbol{f}*}+2L_{\boldsymbol{g}*}^2)t}$$

$$\leq \|\delta \boldsymbol{u}\|^2 (L_{\boldsymbol{f}*} + 2L_{\boldsymbol{g}*}^2)e^{(3L_{\boldsymbol{f}*}+2L_{\boldsymbol{g}*}^2)T}$$

Applying Lemma 13 on 2. and 3. we have that $\boldsymbol{\Phi}_{\boldsymbol{f}*}(\cdot, \cdot, t)$ is $2\left(e^{\frac{2L_{\boldsymbol{f}*}+L_{\boldsymbol{g}*}^2}{2}T} + \sqrt{L_{\boldsymbol{f}*} + 2L_{\boldsymbol{g}*}^2}e^{\frac{3L_{\boldsymbol{f}*}+2L_{\boldsymbol{g}*}^2}{2}T}\right)$–Lipschitz. Applying Lemma 13 on 1. and $\boldsymbol{\Phi}_{\boldsymbol{f}*}(\cdot, \cdot, t)$ and bounding $2 \leq 4$ we finally obtain that $\boldsymbol{\Phi}_{\boldsymbol{f}*}$ is $4\left(e^{\frac{2L_{\boldsymbol{f}*}+L_{\boldsymbol{g}*}^2}{2}T} + \sqrt{L_{\boldsymbol{f}*} + 2L_{\boldsymbol{g}*}^2}e^{\frac{3L_{\boldsymbol{f}*}+2L_{\boldsymbol{g}*}^2}{2}T} + F\right)$–Lipschitz.

$\square$

**Corollary 15** (Lipschitzness of the $\Phi_{b^*}$)**.** *The cost flow $\Phi_{b^*}$ is $\mathcal{O}\left(e^{C_1(L_{\boldsymbol{f}*}+L_{\boldsymbol{g}*}^2)T}\right)$–Lipschitz, where $C_1$ is a constant.*

*Proof.* Same as in the proof of Lemma 14 we first show coordinate-wise Lipschitzness.

1. We first show Lipschitzness in time:

$$\left|\Phi_{b^*}(\boldsymbol{x}, \boldsymbol{u}, t) - \Phi_{b^*}(\boldsymbol{x}, \boldsymbol{u}, \widehat{t})\right| = \left|\mathbb{E}\left[\int_{\widehat{t}}^t b^*(\boldsymbol{x}_s, \boldsymbol{u})ds\right]\right|$$

$$\leq \mathbb{E}\left[\int_{\widehat{t}}^t |b^*(\boldsymbol{x}_s, \boldsymbol{u})|ds\right]$$

$$\leq \mathbb{E}\left[B(t - \widehat{t})\right] = B(t - \widehat{t}).$$

2. To obtain Lipschitzness in state observe:

$$\left|\Phi_{b^*}(\boldsymbol{x}, \boldsymbol{u}, t) - \Phi_{b^*}(\widehat{\boldsymbol{x}}, \boldsymbol{u}, t)\right| = \left|\mathbb{E}\left[\int_0^t b^*(\boldsymbol{\Xi}(\boldsymbol{x}, \boldsymbol{u}, s), \boldsymbol{u}) - b^*(\boldsymbol{\Xi}(\widehat{\boldsymbol{x}}, \boldsymbol{u}, s), \boldsymbol{u})ds\right]\right|$$

$$\leq \mathbb{E}\left[\int_0^t |b^*(\boldsymbol{\Xi}(\boldsymbol{x},\boldsymbol{u},s),\boldsymbol{u}) - b^*(\boldsymbol{\Xi}(\widehat{\boldsymbol{x}},\boldsymbol{u},s),\boldsymbol{u})|\, ds\right]$$

$$\leq L_{b^*}\mathbb{E}\left[\int_0^t \|\boldsymbol{\Xi}(\boldsymbol{x},\boldsymbol{u},s) - \boldsymbol{\Xi}(\widehat{\boldsymbol{x}},\boldsymbol{u},s)\|\, ds\right]$$

$$\leq L_{b^*}\int_0^t \sqrt{\mathbb{E}\left[\|\boldsymbol{\Xi}(\boldsymbol{x},\boldsymbol{u},s) - \boldsymbol{\Xi}(\widehat{\boldsymbol{x}},\boldsymbol{u},s)\|^2\right]}\, ds$$

$$\leq L_{b^*}\|\boldsymbol{x} - \widehat{\boldsymbol{x}}\|\int_0^t e^{\frac{2L_{\boldsymbol{f}^*}+L_{\boldsymbol{g}^*}^2}{2}s}\, ds$$

$$= \frac{2L_{b^*}}{2L_{\boldsymbol{f}^*}+L_{\boldsymbol{g}^*}^2}\left(e^{\frac{2L_{\boldsymbol{f}^*}+L_{\boldsymbol{g}^*}^2}{2}t} - 1\right)\|\boldsymbol{x} - \widehat{\boldsymbol{x}}\|$$

3. Finally, for Lipschitzness in action observe:

$$|\Phi_{b^*}(\boldsymbol{x},\boldsymbol{u},t) - \Phi_{b^*}(\boldsymbol{x},\widehat{\boldsymbol{u}},t)| = \left|\mathbb{E}\left[\int_0^t b^*(\boldsymbol{\Xi}(\boldsymbol{x},\boldsymbol{u},s),\boldsymbol{u}) - b^*(\boldsymbol{\Xi}(\boldsymbol{x},\widehat{\boldsymbol{u}},s),\boldsymbol{u})ds\right]\right|$$

$$\leq \mathbb{E}\left[\int_0^t |b^*(\boldsymbol{\Xi}(\boldsymbol{x},\boldsymbol{u},s),\boldsymbol{u}) - b^*(\boldsymbol{\Xi}(\boldsymbol{x},\widehat{\boldsymbol{u}},s),\boldsymbol{u})|\, ds\right]$$

$$\leq L_{b^*}\mathbb{E}\left[\int_0^t \|\boldsymbol{\Xi}(\boldsymbol{x},\boldsymbol{u},s) - \boldsymbol{\Xi}(\boldsymbol{x},\widehat{\boldsymbol{u}},s)\| + \|\boldsymbol{u} - \widehat{\boldsymbol{u}}\|\, ds\right]$$

$$\leq L_{b^*}t\|\boldsymbol{u} - \widehat{\boldsymbol{u}}\| + L_{b^*}\int_0^t \sqrt{\mathbb{E}\left[\|\boldsymbol{\Xi}(\boldsymbol{x},\boldsymbol{u},s) - \boldsymbol{\Xi}(\boldsymbol{x},\widehat{\boldsymbol{u}},s)\|^2\right]}\, ds$$

$$\leq L_{b^*}t\|\boldsymbol{u} - \widehat{\boldsymbol{u}}\| + L_{b^*}\|\boldsymbol{u} - \widehat{\boldsymbol{u}}\|\sqrt{L_{\boldsymbol{f}^*}+2L_{\boldsymbol{g}^*}^2}\int_0^t e^{\frac{3L_{\boldsymbol{f}^*}+2L_{\boldsymbol{g}^*}^2}{2}s}\, ds$$

$$= L_{b^*}\left(t + \frac{2\sqrt{L_{\boldsymbol{f}^*}+2L_{\boldsymbol{g}^*}^2}}{3L_{\boldsymbol{f}^*}+2L_{\boldsymbol{g}^*}^2}\left(e^{\frac{3L_{\boldsymbol{f}^*}+2L_{\boldsymbol{g}^*}^2}{2}t} - 1\right)\right)\|\boldsymbol{u} - \widehat{\boldsymbol{u}}\|$$

$$\leq L_{b^*}\left(T + \frac{2\sqrt{L_{\boldsymbol{f}^*}+2L_{\boldsymbol{g}^*}^2}}{3L_{\boldsymbol{f}^*}+2L_{\boldsymbol{g}^*}^2}\left(e^{\frac{3L_{\boldsymbol{f}^*}+2L_{\boldsymbol{g}^*}^2}{2}T} - 1\right)\right)\|\boldsymbol{u} - \widehat{\boldsymbol{u}}\|$$

Applying Lemma 13 result follows. $\square$

**Corollary 16** (Lipschitzness of $\boldsymbol{\Phi}^*$). *The unknown function $\boldsymbol{\Phi}^*$ is $L_{\boldsymbol{\Phi}} = L_{\boldsymbol{\Phi}_{\boldsymbol{f}}} + L_{\boldsymbol{\Phi}_b} = \mathcal{O}\left(e^{D(L_{\boldsymbol{f}^*}+L_{\boldsymbol{g}^*}^2)T}\right)$–Lipschitz, where $D$ is constant.*

### A.2.3  Regret bound

**Lemma 17** (Per episode regret bound (general sub-Gaussian noise)). *Consider the setting with a bounded number of switches $K$, and let Assumption 1, 3, and Assumption 4 hold. Then, we get with probability at least $1 - \delta$:*

$$V_{\boldsymbol{\pi}_n,\boldsymbol{\Phi}^*}(\boldsymbol{x}_0,T,K) - V_{\boldsymbol{\pi}^*,\boldsymbol{\Phi}^*}(\boldsymbol{x}_0,T,K) \leq$$

$$\leq \mathcal{O}\left(L_{\boldsymbol{\sigma}}^{K-1}\beta_{n-1}^K e^{C(L_{\boldsymbol{f}^*}+L_{\boldsymbol{g}^*}^2)(1+L_{\boldsymbol{\pi}})TK}\mathbb{E}\left[\sum_{k=0}^K \|\boldsymbol{\sigma}_{n-1}(\boldsymbol{x}_{n,k},\boldsymbol{\pi}_n(\boldsymbol{x}_{n,k},t_{n,k},k))\|_2\right]\right)$$

*Proof.* Applying Lemma 5 of Curi et al. (2020) the result follows. $\square$

**Theorem 18.** *Consider the setting with a bounded number of switches $K$, and let Assumption 1, 3, and Assumption 4 hold. Then, we get with probability at least $1 - \delta$:*

$$R_N = \sum_{n=1}^N V_{\boldsymbol{\pi}_n,\boldsymbol{\Phi}^*}(\boldsymbol{x}_0,T,K) - V_{\boldsymbol{\pi}^*,\boldsymbol{\Phi}^*}(\boldsymbol{x}_0,T,K)$$

$$\leq \mathcal{O}\left(L_{\boldsymbol{\sigma}}^{K-1}\beta_{N-1}^{K}\sqrt{K}e^{C(L_{\boldsymbol{f}^*}+L_{\boldsymbol{g}^*}^2)(1+L_{\boldsymbol{\pi}})TK}\sqrt{N\mathcal{I}_N}\right)$$

*Proof.* We apply Lemma 17 and Cauchy-Schwarz:

$$
\begin{aligned}
R_N &= \sum_{n=1}^{N} V_{\boldsymbol{\pi}_n, \boldsymbol{\Phi}^*}(\boldsymbol{x}_0, T, K) - V_{\boldsymbol{\pi}^*, \boldsymbol{\Phi}^*}(\boldsymbol{x}_0, T, K) \\
&\leq \sum_{n=1}^{N} \mathcal{O}\left(L_{\boldsymbol{\sigma}}^{K-1}\beta_{n-1}^{K}e^{C(L_{\boldsymbol{f}^*}+L_{\boldsymbol{g}^*}^2)(1+L_{\boldsymbol{\pi}})TK}\mathbb{E}\left[\sum_{k=0}^{K}\|\boldsymbol{\sigma}_{n-1}(\boldsymbol{x}_{n,k}, \boldsymbol{\pi}_n(\boldsymbol{x}_{n,k}, t_{n,k}, k))\|_2\right]\right) \\
&\leq \mathcal{O}\left(L_{\boldsymbol{\sigma}}^{K-1}\beta_{N-1}^{K}e^{C(L_{\boldsymbol{f}^*}+L_{\boldsymbol{g}^*}^2)(1+L_{\boldsymbol{\pi}})TK}\right)\mathbb{E}\left[\sum_{n=1}^{N}\sum_{k=0}^{K}\|\boldsymbol{\sigma}_{n-1}(\boldsymbol{x}_{n,k}, \boldsymbol{\pi}_n(\boldsymbol{x}_{n,k}, t_{n,k}, k))\|_2\right] \\
&\leq \mathcal{O}\left(L_{\boldsymbol{\sigma}}^{K-1}\beta_{N-1}^{K}e^{C(L_{\boldsymbol{f}^*}+L_{\boldsymbol{g}^*}^2)(1+L_{\boldsymbol{\pi}})TK}\right)\sqrt{K}\sqrt{N\mathcal{I}_N}
\end{aligned}
$$

Here we first applied Lemma 17. Then we used the monotonicity of $(\beta_n)_{n\geq 0}$ sequence. In the last step we first applied maximum over the collected data, then Cauchy-Schwarz inequality and finally the definition of model complexity. $\square$

# B    Additional Experiments

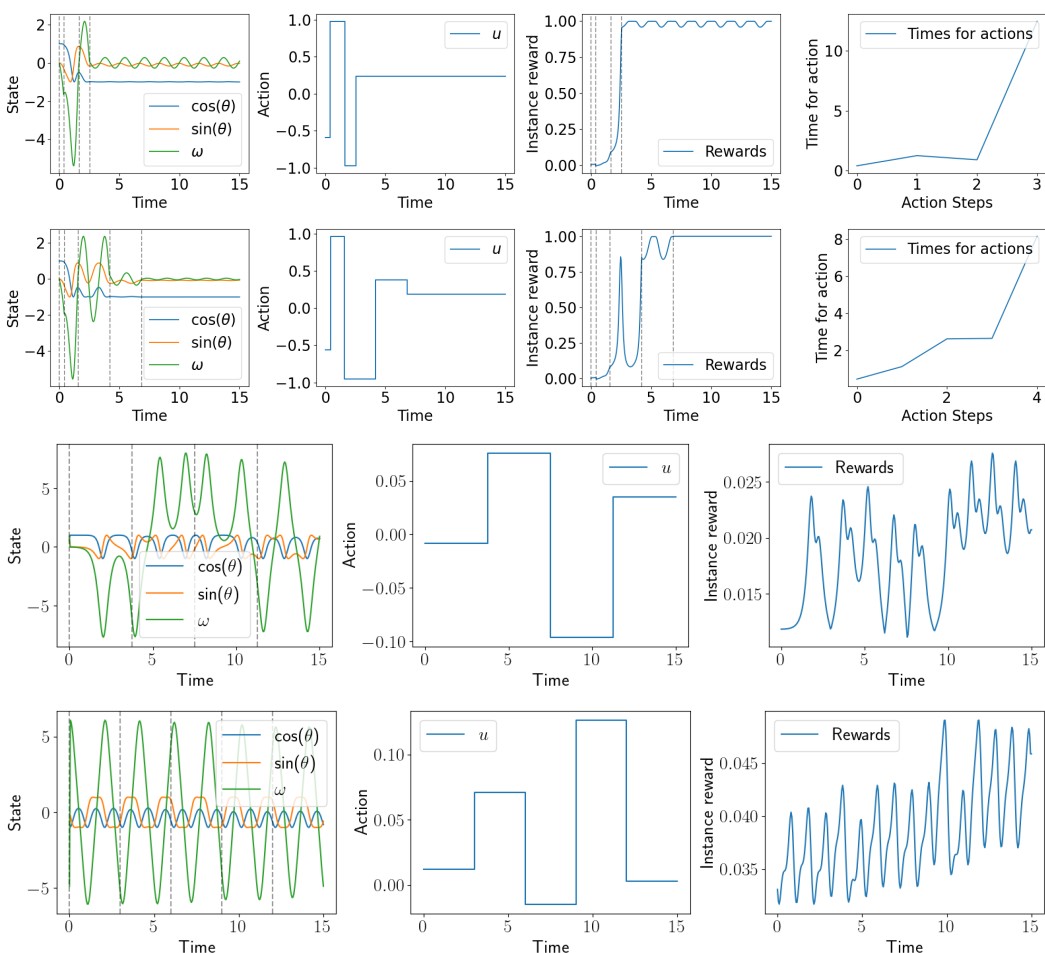

Figure 6: Pendulum swing-down task. Row 1: 4 interactions, optimized interaction times, Row 2: 5 interactions, optimized interaction times, Row 3: 4 equidistant interactions, Row 4: 4 equidistant interactions.

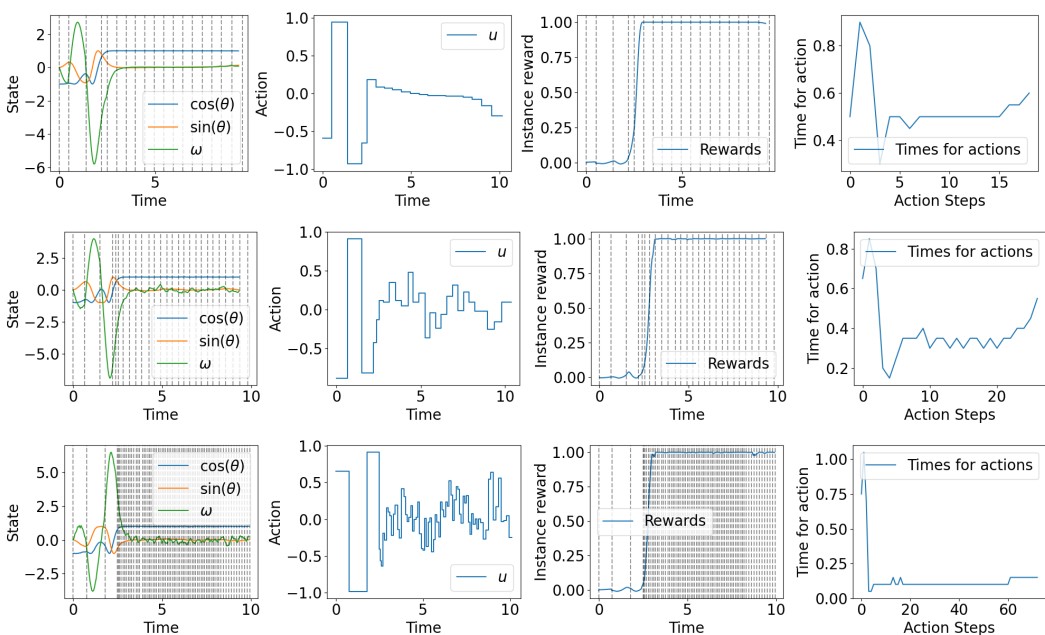

Figure 7: When stochasticity of the environments increases, we need more interactions at the unstable equilibrium (Pendulum on top). The stochasticity scale goes from 0.1 to 0.5 to 1.0 from top to bottom row respectively.

