# OpenReview forum: "When to Sense and Control? A Time-adaptive Approach for Continuous-Time RL"
_NeurIPS.cc/2024/Conference — NeurIPS 2024 poster_

### Official Review · Reviewer_ZHjB · 2024-06-13

**Soundness:** 3
**Presentation:** 2
**Contribution:** 2
**Rating:** 6
**Confidence:** 3

**Summary:**

The paper focuses on continuous-time reinforcement learning. Typically the control and the measurements of continuous-time systems occur at discrete time points. These interventions generally come with a cost. The paper proposes a time-adaptive approach to reduce the number of these interventions in an optimal manner. Therefore they formulate the problem as an MDP, for which standard RL algorithms can be used. Additionally the authors propose a model-based algorithm for their setting.

**Strengths:**

1. The paper describes an interesting setting. The problem of choosing the optimal measurement and control time-points for continuous-time systems is relevant for many processes in nature and industry.

2. The reformulation of the original problems to discrete-time MDPs is well described.

3. The various experiments visualize and describe the effects of the hyperparameters nicely.

**Weaknesses:**

1. The references in the paper lack consistency in formatting. Conference names are sometimes written in full, sometimes abbreviated, and occasionally omitted altogether.

2. One major issue is the lack of details on the learning process. It would be interesting to compare the learning curves of the equidistant and the time-adaptive approach.

3. The experimental section leaves some questions open -> See Questions


Minor Weaknesses:
- Figure 3 comes before Figure 2
- Page 5, Line 134 says time discretization splits the whole horizon in T/K discrete time points. I think it should be K time points with length T/K
- It is difficult to discern the decline in interactions for the greenhouse task in Figure 3 first row, which coincides with the drop in episode reward.

**Questions:**

1. Regarding the experiments for Figure 2: After what time fraction are the greenhouse and pendulum swing-down usually in the stable equilibrium? As described, if the systems are in the stable equilibrium, the discretized approach continues with interactions, while the time-adaptive approach needs less. It would be interesting to compare the interactions for the time until the stable equilibrium is reached.

2. Figure 2, Pendulum Swing-up: Why does the equidistant approach perform so poorly? In Figure 1, the task could be solved with K=5 number of interactions, without high-frequency changes. Intuitively the equidistant approach should perform well with more than 10 interactions.

3. How many iterations were used for learning the equidistant policy and the SAC-TaCos policy in Figure 2? It appears that learning the equidistant policy should be easier, as it has smaller input and output spaces.

4. In the first row of Figure 3, the interaction cost increases, yet the reward does not decline significantly. It would be interesting to identify the point at which the reward starts to drop.

**Limitations:**

Limitations have been adressed adequately.

---

> ### Author Rebuttal · Authors · 2024-08-05
>
> Thank you for your comments and valuable feedback!
>
> # Weaknesses
>
> 1. *“The references in the paper lack consistency in formatting. Conference names are sometimes written in full, sometimes abbreviated, and occasionally omitted altogether.”:* Thanks for spotting that, we updated the references in the updated version of the paper.
>
> 2. *“One major issue is the lack of details on the learning process. It would be interesting to compare the learning curves of the equidistant and the time-adaptive approach.”:* We generally observed the trend that when we use our algorithm in a model-free fashion the learning is faster compared to the standard training. This phenomenon has been observed in several other works as well [1, 2]. We also provide learning curves in the bottom row of Figure 4.
>
> 3. *“Figure 3 comes before Figure 2”:* Thanks for spotting that, we fixed the ordering in the updated version of the paper.
>
> 4. *“Page 5, Line 134 says time discretization splits the whole horizon in T/K discrete time points. I think it should be K time points with length T/K”:* Indeed, thanks for spotting that. We changed it in the updated version of the paper.
>
> 5. *“It is difficult to discern the decline in interactions for the greenhouse task in Figure 3 first row, which coincides with the drop in episode reward.”:* In this experiment the change in the dynamics is very slow, hence only few control changes over the entire horizon of 25h suffice. As soon as there is a certain cost for the interaction, the number of interactions needed significantly drops (you can see the drop on the left side of the plot), after the initial drop the number of interactions stays more or less the same with increasing cost, since interactions are already at the minimum of what is needed to successfully track the temperature.
>
> # Questions
>
> 1. *“Regarding the experiments for Figure 2: After what time fraction are the greenhouse and pendulum swing-down usually in the stable equilibrium? As described, if the systems are in a stable equilibrium, the discretized approach continues with interactions, while the time-adaptive approach needs less. It would be interesting to compare the interactions for the time until the stable equilibrium is reached.”:* The Greenhouse and Pendulum swing-down systems we use in the experiment for Figure 2 preserve the energy (no friction or similar losses of energy) so technically they never reach exactly (due to numerical errors) the stable equilibrium and only reach its neighborhood. This is best illustrated with the first 2 rows of Figure 7 in the pdf attached to the general response, where we plot the trajectories of the Pendulum swing-down task. We show the trajectories with 4 and 5 interactions. With 4 interactions only, we can drive the system close to the stable equilibrium, while with 5 interactions we can drive it even closer.
> At the same time, if we try to solve the task with equidistant control, the control frequency is too low, and with only 4 or 5 interactions over the horizon 15 seconds, we can not push the system close to the equilibrium, as can be seen in the bottom 2 rows of Figure 7 in the pdf attached to the general response.
>
> 2. *“Figure 2, Pendulum Swing-up: Why does the equidistant approach perform so poorly? In Figure 1, the task could be solved with K=5 number of interactions, without high-frequency changes. Intuitively the equidistant approach should perform well with more than 10 interactions.”:* Comparison between the setting in Figure 1 and equidistant in Figure 2 is not as straightforward as it may seem at first. The first thing is that in Figure 1 we used a shorter horizon, i.e. 5 seconds compared to 10 seconds in Figure 2, which means that the same number of actions in Figure 2 means that each action lasts double the time. The second thing is that in Figure 1 we optimized the number of times between interactions, which results in some interactions taking 0.8 seconds and some interactions 1.3 seconds. With only equidistant interactions we need considerably more interactions to account for not being able to optimize the interaction times.
>
> 3. *“How many iterations were used for learning the equidistant policy and the SAC-TaCos policy in Figure 2? It appears that learning the equidistant policy should be easier, as it has smaller input and output spaces.”:* For training in Figure 2 we used 200000 env interactions with 200000 gradient steps for SAC-TACOS. For equidistant training, we used the same amount of observations with the same amount of gradient updates. This results also in an equal amount of physical interaction time when comparing SAC-TACOS and equidistant training. Perhaps surprisingly, training with an additional degree of freedom, where we can choose the length of actions results in faster convergence as also observed in other works with similar settings [1, 2]. In our case, we choose 200000 iterations to make sure that both algorithms converge (for Pendulum swing-up and swing-down, we could in many cases obtain the same performance with only 20000 iterations).
>
> 4. *“In the first row of Figure 3, the interaction cost increases, yet the reward does not decline significantly. It would be interesting to identify the point at which the reward starts to drop.”:* Thanks for pointing this out, we extended our experiments to cover that case as well. Look at Figure 8 in the pdf attached to the general response.
>
> [1] Ni, Tianwei, and Eric Jang. "Continuous control on time." ICLR 2022 Workshop on Generalizable Policy Learning in Physical World. 2022.
>
> [2] Biedenkapp, André, et al. "TempoRL: Learning when to act." International Conference on Machine Learning. PMLR, 2021.
>
> Having addressed the reviewer’s concerns, we would appreciate if they could reevaluate their assessment of our work. We are happy to address any additional questions.

---

> > ### Author Response · Authors · 2024-08-09
> > **Follow up on rebuttal**
> >
> > Dear Reviewer,
> >
> > We hope this message finds you well. We have noticed that our detailed rebuttal, addressing each of the concerns raised in your review, has not yet received any feedback. We understand the demanding nature of the review process and appreciate the time and effort invested in evaluating our work.
> >
> > We kindly urge you to consider our responses to your questions, as we believe they adeptly address your concerns. With some days left, we hope we can still have a fruitful rebuttal period.
> >
> > Thank you,
> > Authors

---

> > ### Comment · Reviewer_ZHjB · 2024-08-12
> >
> > Thank you for adressing my concerns. I appreciate the novelty of the work and therefore increase my score accordingly.

---

> > > ### Author Response · Authors · 2024-08-13
> > > **Official Comment**
> > >
> > > Thanks for raising our score and the active engagement in the review process!

---

### Official Review · Reviewer_BvQD · 2024-07-10

**Soundness:** 3
**Presentation:** 3
**Contribution:** 2
**Rating:** 6
**Confidence:** 4

**Summary:**

The paper introduces a framework for reinforcement learning named Time-adaptive Control & Sensing (TACOS). The TACOS framework reformulates the problem of continuous-time RL into an equivalent discrete-time Markov decision process (MDP) that standard RL algorithms can solve. Additionally, a model-based version of TACOS, named OTACOS, is proposed to reduce the sample complexity.

**Strengths:**

1. The introduction of the Time-adaptive Control & Sensing (TACOS) framework is a novel approach that creatively combines the challenges of continuous-time dynamics with the necessity of minimizing interactions due to cost considerations.
2. The paper provides a strong theoretical foundation for the TACOS framework, with clear reformulation of continuous-time RL problems into an equivalent discrete-time MDP.
3. The empirical results support the theoretical claims, with demonstrations of TACOS and OTACOS outperforming traditional discrete-time counterparts

**Weaknesses:**

1. Although the problem posed by the paper is novel and interesting, the solution simply involves incorporating time $t$ into the action space for learning, which a little lacks novelty.
2. The empirical validation is somewhat limited in diversity, primarily focusing on controlled synthetic environments. This limitation might affect the generalizability of the results to more complex or noisy real-world systems.

**Questions:**

The model learned here differs from the typical models in model-based reinforcement learning, as it incorporates time as a transition variable, making the learning process more challenging. Could the authors demonstrate the learning performance of the model itself?

**Limitations:**

The paper does not show a discussion of limitations. Authors could enhance their discussion by explicitly addressing the potential limitations in real-world scenarios that involve complex, noisy, or non-stationary environments.

---

> ### Author Rebuttal · Authors · 2024-08-05
>
> First, thank you a lot for your positive and valuable feedback! We will indeed incorporate the proposed ideas in the updated version of the paper.
>
> # Weaknesses
>
> 1. *“Although the problem posed by the paper is novel and interesting, the solution simply involves incorporating time into the action space for learning, which a little lacks novelty.”:* We agree that augmenting the actions with time is very simple, and we see it rather as positive, since with little modification we can obtain an algorithm that hasn’t been extensively studied before and performs very well on standard deep RL benchmarks, outperforming the classical RL algorithms such as SAC and PPO.
>
> 2. *“The empirical validation is somewhat limited in diversity, primarily focusing on controlled synthetic environments. This limitation might affect the generalizability of the results to more complex or noisy real-world systems.”:* We acknowledge that our algorithm has been tested only on synthetic environments and not on real-world systems. While implementation of the algorithm on real-world systems (e.g. robotic hardware) is a challenge on its own, we tried to showcase and evaluate our algorithm in simulation extensively. We tested the influence of the interaction cost on the algorithm, we showed the influence of requiring as little interactions as possible. We also studied the effects of stochasticity on our algorithm’s performance (c.f., Figure 3). Regarding the complexity of the systems, we tested our algorithm on the humanoid environment, where the dimensions of state and action space are 244 and 17 respectively and the dynamics are highly non-linear. The system is, to the best of our knowledge, considered as quite complex system in the community.
>
> # Questions
>
> 1. *“The model learned here differs from the typical models in model-based reinforcement learning, as it incorporates time as a transition variable, making the learning process more challenging. Could the authors demonstrate the learning performance of the model itself?”*
> We agree, learning the dynamics function that adds to the standard state and action also a time component as an input is more challenging. We think the best assessment of how good the model is, is whether the model-based RL algorithm converges, and in our case we can see that with an increasing number of episodes, the controller that uses the learned model obtaines higher and higher rewards on the true system, eventually converging to the best attainable reward. For example, in the Pendulum swing-up task the model is good enough to solve the task almost perfectly after only 8 episodes.
>
> # Limitations
>
> 1. *“The paper does not show a discussion of limitations. Authors could enhance their discussion by explicitly addressing the potential limitations in real-world scenarios that involve complex, noisy, or non-stationary environments.”:* We added in the conclusion of the updated version of the paper that our algorithm has been only tested in the simulation and that we leave testing the algorithm on real-world environments for exciting future work.
>
> Having addressed the reviewer’s concerns, we would appreciate if the reviewer would consider reevaluating their score for our work.

---

> > ### Author Response · Authors · 2024-08-09
> > **Follow up on rebuttal**
> >
> > Dear Reviewer,
> >
> > We hope this message finds you well. We have noticed that our detailed rebuttal, addressing each of the concerns raised in your review, has not yet received any feedback. We understand the demanding nature of the review process and appreciate the time and effort invested in evaluating our work.
> >
> > We kindly urge you to consider our responses to your questions, as we believe they adeptly address your concerns. With some days left, we hope we can still have a fruitful rebuttal period.
> >
> > Thank you,
> >
> > Authors

---

> > ### Comment · Reviewer_BvQD · 2024-08-10
> >
> > Thank you for your replay!
> >
> > The significant improvements achieved by the algorithm in these environments is impressive, and I also recognize the novelty of the paper. I raised my score accordingly. The reason I questioned the complexity of the environment is because I think that while it is easy to understand improvements in some environments after adding the time adaptive control framework. Once the environment becomes more complex, the increased decision complexity could offset the gains in performance, or even lead to worse performance. This concern makes me worried about the scalability of the algorithm. That's why I didn't give a higher score.

---

> > > ### Author Response · Authors · 2024-08-10
> > > **Response to reviewer BvQD**
> > >
> > > Thanks for your active engagement during the rebuttals, for acknowledging our contribution, and for increasing our score.
> > >
> > > We evaluated our algorithm on well-established deep RL benchmarks with varying complexity/dimensionality (c.f., Figure 4). Particularly, we also tested our algorithm on the humanoid environment from the OpenAI gym (244 dimensional state, 17 dimensional input) to demonstrate the scalability of TACOS.
> > > In all the environments, we see a clear benefit of the TACOS framework.  We hope this also convinces the reviewer regarding the scalability of our approach.

---

### Official Review · Reviewer_L1VF · 2024-07-13

**Soundness:** 3
**Presentation:** 3
**Contribution:** 3
**Rating:** 8
**Confidence:** 3

**Summary:**

This paper proposes a novel time-adaptive RL method framework (TaCoS) for continuous-time systems with continuous state and action spaces. The framework shows that the settings of interactions having costs or a budget of interactions, can be formulated as extended MDPs, that can be solved with standard RL algorithms. The paper theoretically demonstrates this, and empirically verifies this. The paper also empirically demonstrates that TaCoS works across a range of interaction frequencies, and proposes OTaCoS a model-based RL approach.

**Strengths:**

* The paper is well written, and the intuitive explanations help the reader.
* The problem of time-adaptive RL for continuous-time systems with continuous state and action spaces is of high significance to the community and is well-motived throughout.
* The paper contributions are novel to the framework proposed by TaCoS and provide theoretical and empirical evidence.
* The method works surprisingly well, with the surprise that TaCoS achieves better control performance even with a small budget of interactions.

**Weaknesses:**

* L154: “Intuitively, the more stochastic the environment, the more interactions we would require to stabilize the system.” the argument should either have a reference or be empirically verified.
* Minor: Missing related work reference of (Nam et al. 2021).

References:

* Nam, HyunJi Alex, Scott Fleming, and Emma Brunskill. "Reinforcement learning with state observation costs in action-contingent noiselessly observable Markov decision processes." Advances in Neural Information Processing Systems 34 (2021): 15650-15666.

**Questions:**

* Is the action applied from the policy kept constant until the next interaction? I presume the action is kept constant. Have you considered parameterizing the action and enabling non-constant actions or even continuous trajectories of actions between interaction times as a form of open-loop planning?

**Limitations:**

Yes, they are discussed in Section 5.

---

> ### Author Rebuttal · Authors · 2024-08-05
>
> Thanks a lot for the positive and valuable feedback!
>
> # Weaknesses
>
> 1.  *Verification of “Intuitively, the more stochastic the environment, the more interactions we would require to stabilize the system.”:* We verify this empirically. In Figure 3 in the paper, bottom row, we analyze the setting where we increase the stochasticity of the environment. With larger stochasticity, the number of interactions increases (in the Pendulum swing-up task the number of interactions increases from 20 to 60). Does that answer your question? We also provide these 3 additional Figures where we plot trajectories at environment stochasticity scale 0.1, 0.5, and 1.0 for further intuition (look at Figure 6 in the pdf attached to the general response). We included these Figures in the appendix of the updated version of the paper.
>
> 2. *Additional related work:* Thanks for pointing out to additional related work, we added it to the updated version of the paper.
>
> # Questions
>
> 1. *Action between the interactions:* In all the experiments we conducted the action is kept constant between the two interactions. The case when we allow for open-loop planning between the interactions indeed makes a lot of sense and is very interesting. We leave it as an exciting future work.
>
> We hope we addressed your concerns and questions. Please reach out if you need additional clarifications.

---

> > ### Comment · Reviewer_L1VF · 2024-08-12
> >
> > Thank you for addressing my concerns and questions. I have raised my score.

---

> > > ### Author Response · Authors · 2024-08-13
> > > **Official Comment**
> > >
> > > Thanks for raising our score and the active engagement in the review process!

---

### Official Review · Reviewer_XqGZ · 2024-07-30

**Soundness:** 3
**Presentation:** 3
**Contribution:** 3
**Rating:** 7
**Confidence:** 3

**Summary:**

Reinforcement learning (RL) is effective for discrete-time Markov decision processes (MDPs), but many systems operate continuously in time, making discrete-time MDPs less accurate. In applications like greenhouse control or medical treatments, each interaction is costly due to manual intervention. To address this, we propose a time-adaptive approach, Time-adaptive Control & Sensing (TACOS), which optimizes policies that also predict the duration of their application. This results in an extended MDP solvable by standard RL algorithms. TACOS significantly reduces interactions while maintaining or improving performance and robustness. We also introduce OTACOS, a model-based algorithm with sublinear regret for smooth dynamic systems, offering further sample-efficiency gains.

**Strengths:**

- handling continuosu control is sort ot interesting since this also bridge the gap between real-world RL and RL in simulation.

**Weaknesses:**

My questions are listed as below

**Questions:**

Thank you for the opportunity to review your paper. I have a few questions and observations outlined below.

1. Real-world applications and global frequencies.

- The discussion around the requirement for different global frequencies in real-world applications is intriguing (line 32), as it highlights the limitations of discrete-time sampling (lines 30-32). However, it seems this limitation is not fully addressed by continuous-time control (or the TACOS algorithm). The fundamental difference between discrete-time control and continuous-time control lies in the amount of environmental information included to compute the policy. For instance, within the total time interval $[0,T]$, discrete-time control computes the policy based on $K$ sampled data points {$t_1, t_2, \cdots, t_K$}, resulting in an optimal policy only for those specific sampled times. In contrast, continuous-time control computes an optimal policy over the entire continuous duration $[0,T]$ and then applies it at discrete times (constrained by Equations (2) and (3)). Utilizing the policy derived from a continuous-time formulation in a discrete-time setting would provide better generalization ability to obtain higher rewards for $t \notin$ {$ t_1,t_2,\cdots,t_K$} but ***does not necessarily address the limitation mentioned in lines 30-32.***

2. Definition of Notations:

- The paper lacks precise definitions of certain notations. For example, the policy $\pi: \mathcal{X} \to \mathcal{U}$ maps states to control inputs, but the exact meaning of $\pi_{\mathcal{T}}$ is unclear. In line 80, it is stated that "$\pi_{\mathcal{T}}$ is a policy that predicts the duration of applying the action." This suggests $\pi_{\mathcal{T}}$ is a prediction variable rather than a policy, as the term "policy" typically refers to decisions or actions taken. Additionally, while $t_{i} = \pi_{\mathcal{T}}(x_{t_{i-1}}) + t_{i-1}$ is mentioned, the role of $\pi_{\mathcal{T}}$ in the objective function needs further clarification.

Also have some minor issues..

- Line 65-66 and 68-69 discuss "real-time inference" or "adaptive control approach," which is a known challenge in real-world reinforcement learning. Including references to related literature would help readers better understand these contributions as follows.

[1] Dulac-Arnold, G., Mankowitz, D., and Hester, T.Challenges of real-world reinforcement learning.arXiv preprint arXiv:1904.12901, 2019.

[2] Hyunin Lee, Ming Jin, Javad Lavaei, Somayeh Sojoudi, Pausing Policy Learning in Non-stationary Reinforcement Learning. ICML 2024

[3] Al-Shedivat, M., Bansal, T., Burda, Y., Sutskever, I., Mordatch, I., and Abbeel, P.Continuous adaptation via meta-learning in nonstationary and competitive environments.In ICLR, 2018.

- Also please discuss what SDE stands for.

**Limitations:**

No potential negative societal impact.

---

> ### Author Rebuttal · Authors · 2024-08-05
>
> Thank you for your comments and valuable feedback!
>
> # Questions
>
> 1. *Real-world applications and global frequencies*: As you correctly described, the main difference between standard discrete-time control and our setting is that in the standard discrete-time setting, the interaction (measuring the system and changing the controller) happens only at equidistant points in time.  Our approach, also optimizes over the interaction times, that is our algorithm suggests times $t_1, \dots, t_K$ at which we should exhibit control. In the case of the doctor and patient, the algorithm can then adapt interaction times such that the interaction frequency is not global (equidistant) but adapts to the patient's needs hence solving the problem we mentioned in lines 30-32. A good example of why this is the case are first 2 rows of Figure 7 in the pdf attached to the general response. There the task is to bring the Pendulum to stable equilibrium. We do several interactions in the beginning when the Pendulum is out of stable equilibrium (think of a sick patient) and don't do many interactions when the Pendulum is in stable equilibrium (think of a healthy patient).
>
>
> 2. *Definition of notations:* We are happy to improve our notation. We agree, when we defined the policies, we were not consistent. We changed the notation in the update version of the paper as follows: The part of the policy that maps to the action space $\mathcal{U}$ we denote as $\pi_{\mathcal{U}}$ and the part of the policy that maps into the time for how long we apply the policy we denote as $\pi_{\mathcal{T}}$. When the two parts of the policy are concatenated we write only $\pi = (\pi_{\mathcal{U}}, \pi_{\mathcal{T}})$.
>
> 3. *Additional related work:* Thanks for pointing out the additional literature that would help the reader to better understand the contributions. We added them in the updated version of the paper.
>
> 4. *What does SDE stand for?* We already explained the acronym SDE in lines 22-23 of the paper. It stands for the Stochastic Differential Equation.
>
> Having addressed your concerns,  we kindly ask you to consider revising our score. For any remaining questions, we are happy to provide further clarification.

---

> > ### Comment · Reviewer_XqGZ · 2024-08-09
> >
> > Dear Authors,
> >
> > Thank you for the update.
> >
> > ***Question 1:*** I appreciate the direction you're aiming for with your contribution. However, I still have some concerns that I'd like to clarify. To ensure I understand correctly: In continuous-time control, we have interaction times  $t \in $ {$ t_1, t_2, \ldots, t_K$}. This means that the action executed at time $t_2$ fully considers the duration between $t_2 $and $t_3$. However, if the system requires a different interaction frequency and needs to take action at any time $t^\prime \in [t_2, t_3]$, how does the continuous-time control compute the action at time $t^\prime$?
> >
> > - If the action is indeed a function of time $t$, it makes sense that the action at time $t^\prime$ is computable. However, in this case, how is this method superior to simply interpolating between the actions executed at $t_2$ and $t_3$ in "discrete time sampling"?
> >
> > ***Question 2:*** Thank you. This is fully addressed. Please use that notation in the revised version if accepted.
> >
> > ***Question 3:*** Thank you. This is fully addressed. Please add the reference in the revised version if accepted.
> >
> > ***Question 4:*** Thank you. This is also fully addressed.

---

> > > ### Author Response · Authors · 2024-08-09
> > > **Response to reviewer's comments**
> > >
> > > Thanks for actively participating in the review process. We are glad that we could address most of your concerns. Below is our response to your question 1.
> > >
> > > **Answer to Q1**: We would like to clarify that our framework, unlike standard RL, allows the policy to choose the interaction times $t_{1}, \dots, t_{k}$. Moreover, the algorithm works in the following way;
> > >
> > > At $t_2$, the policy outputs based on the state $s_{t_2}$ the action $a_{t_2}$ and how long it should be applied, i.e., $t_3 - t_2$.  In our experiments, this action is kept constant for the whole duration (zero order hold). Hence, the action remains the same between $[t_{2}, t_{3})$. Our framework allows the policy to be adaptive, i.e., the policy adaptively selects the frequency of control based on the state, rather than having a constant frequency (standard discrete-time control).
> > > In a typical RL fashion, the policy is optimized by directly interacting with the system and it tries to maximize the expected reward.  Therefore, it knows what frequency of control the system requires at the state $s_{t_2}$ from interactions. If the system is very noisy, intuitively you would expect the policy to exhibit more control. This is also observed in our experiments (c.f., Figure 3 in the main paper and Figure 6 in the PDF with additional experiments).
> > >
> > > Our method is better than standard discrete time control since our policy is adaptively selecting the times at which control is exhibited ($t_2$, $t_3$, etc). In discrete time this is specified by the problem designer ($t_i^{discrete} \neq t_i^{TACOS}$) which might be suboptimal. This is also what we show in our experiments (c.f., Figure 4 in the paper and Figure 7, 8 in the additional experiments).
> > >
> > > Furthermore, while in all our experiments we focus on zero-order control, one could consider a different control strategy, e.g., linear interpolation w.r.t. time as suggested by the reviewer. We stuck with zero-order control cause of its simplicity and because it performed well for all experiments.
> > >
> > > We hope this addresses your concerns and kindly ask you to consider revising our score. For any remaining questions, we are happy to provide further clarification.

---

> > > > ### Comment · Reviewer_XqGZ · 2024-08-10
> > > >
> > > > Dear Author,
> > > >
> > > > Thank you for the clarification. Question 1 is almost fully addressed -- The reviewer just has one final question regarding how to determine the interaction time sequence {$t_1, t_2, \dots, t_K$} and the number of interactions $K$.
> > > >
> > > > **I am inclined to support the acceptance of this paper. However, before I raise my score, I would like to kindly ask the author to clearly highlight the contributions of this work.** It appears that my Question 1 is more closely related to the advantages of continuous control over discrete-time control. It would be beneficial to understand the specific contributions of this work, whether it be in the formulation, experimental results, or another aspect -- please feel free to be creative in your explanation -- particularly in comparison with other continuous-time reinforcement learning algorithms.
> > > >
> > > > Thank you.

---

> > > > > ### Author Response · Authors · 2024-08-10
> > > > > **Response to Reviewer XgGZ**
> > > > >
> > > > > Thanks again for your questions and active participation in the rebuttal period.
> > > > >
> > > > > The time $t_0$ is typically $0$ (initial time). Next, based on the state $s_0$ (which includes the time to-go), the policy outputs an action $a_0$ and how long it should be applied $\Delta t_0$. Therefore, $t_1 = t_0 + \Delta t_0$. Then at $t_1$ the state is measured again, the action $a_1$ and $\Delta t_1$ are returned by the policy and we get $t_2 = t_1 + \Delta t_1$. This process is then repeated at $t_2$ and so on.
> > > > >
> > > > > How does the policy decide on $\Delta t$? In the setting with interaction costs (section 3.1), there is a cost every time the system changes its control/measures the system. This incentivizes the policy to have large values for $\Delta t$ (keep the same action for as long as possible -> change the input as few times as possible/minimize interactions). However, the policy has to also maximize the reward and thus it cannot pick arbitrarily high values for $\Delta t$. In a typical RL fashion, the policy learns to trade off these two objectives.
> > > > >
> > > > > Our formulation, TACOS, shows that such problems can be solved by any off-the-shelf RL algorithm (section 3.1 and section 3.2). Moreover, our contributions are as follows:
> > > > >
> > > > > 1) TACOS reformulates the time-adaptive RL setting (where policy decides when to interact with the system) to a standard discrete-time MDP for which any RL algorithm can be used (section 3.1 and section 3.2).
> > > > >
> > > > > 2) In our experiments, we show that sota model-free RL methods such as SAC and PPO can be applied off-the-shelf with the TACOS reformulation and perform much better than the standard discrete-time control setting, i.e., we show the benefits of time-adaptive RL and also that typical RL methods can be simply used for this setting.
> > > > >
> > > > > 3) We also show that for the time-adaptive setting, the regret bounds from mode-based RL for the discrete-time setting are also retained (section 5, theorem 2).
> > > > >
> > > > > In summary, our reformulation (TACOS) allows us to study time-adaptive RL problems with our standard and well-developed discrete-time RL toolset.  Further, the TACOS framework performs much better than discrete-time RL, while enjoying the same guarantees. We will further clarify this in the paper. Hope this addresses the concern of the reviewer. We'd appreciate the reviewer revising our score.

---

> > > > > > ### Comment · Reviewer_XqGZ · 2024-08-10
> > > > > >
> > > > > > Thanks for the update.
> > > > > >
> > > > > > My questions are fully addressed -- **I have raised my score to 7 and stands on the accept side** -- contributions are reasonable. Thanks for the active engagement. **Please revise the paper based on four questions** if the paper gets accepted. Good luck!

---

> > > > > > > ### Author Response · Authors · 2024-08-10
> > > > > > > **Acknowledging reviewer’s active engagement**
> > > > > > >
> > > > > > > Thanks for your active engagement in the rebuttal period and for increasing our score. We will revise the paper based on your feedback as suggested.

---

### Author Rebuttal · Authors · 2024-08-05

We thank all the reviewers for their valuable and useful feedback. We have attached a pdf with 3 additional experiments that study the following setting:
1. *Figure 6:* We further show the effect of environment stochasticity on the frequency of interactions. We empirically show that with larger stochasticity the algorithm predicts interactions that last a shorter amount of time, on the Pendulum swing-up task.
2. *Figure 7:* We illustrated the difference between our algorithm and the algorithm with equidistant interactions around stable equilibrium in the Pendulum swing-down task. While our algorithm adapts and makes fast interactions at the beginning to bring the Pendulum down to the stable equilibrium and then doesn't interact with the system anymore, the algorithm with equidistant interactions can't adapt interactions time and can't even bring the systems to the stable equilibrium in one episode time.
3. *Figure 8:* We extended part of the Figure 3 from the paper to the case with higher interaction cost so that the Reward [Without Interaction Cost] starts decreasing.

We included the results of the additional experiment in the appendix of the updated version of the paper. We would be happy to further update the paper if there are any remaining questions or feedback from the reviewers.

---

### Decision · Program_Chairs · 2024-09-25

**Decision:**

Accept (poster)

**Comment:**

The reviewers found the work novel, well-written, and well-supported by experiments. They all had questions that were properly answered by the authors, thus, they all raised their scores during the rebuttals. It would be great if the authors revise the paper to include these answers that properly addressed the reviewers' questions.